# Sensitivity of a Sahelian groundwater-based agroforestry system to tree density and water availability using the land surface model ORCHIDEE (r7949)

Espoir Koudjo Gaglo[1,2], Emeline Chaste[5], Sebastiaan Luyssaert[7], Olivier Roupsard[2,4,5], Christophe Jourdan[5], Sidy Sow[2,3], Nadeige Vandewalle[8,9], Frédéric C. Do[2,5,6], Daouda Ngom[1], Aude Valade[5]

[1]Université Cheikh Anta Diop, Dakar, Senegal
[2]LMI IESOL, IRD, ISRA, Bel Air, Dakar, Senegal
[3]Université Gaston Berger, Saint-Louis, Senegal
[4]CIRAD, UMR Eco&Sols, Dakar, Senegal
[5]Eco&Sols, University Montpellier, CIRAD, INRAE, Institut Agro, IRD, Montpellier, France
[6]IRD, UMR Eco & Sols, Univ Montpellier, IRD, CIRAD, INRAE, Montpellier, France
[7]A-LIFE, Vrije Universiteit Amsterdam, Amsterdam, Netherlands
[8]Earth and Life Institute, Faculty of Bioscience Engineering, Catholic University of Louvain, Louvain-la-Neuve, Belgium
[9]Terra Teaching and Research Center, University of Liège, Gembloux Agro-Bio Tech, Gembloux, Belgium

*Correspondence:* Espoir Koudjo Gaglo, (espoirkoudjo.gaglo@ucad.edu.sn)

**Abstract.** The Sahel region is characterized by its semi-arid climate and open-canopy agroforestry systems, which play an important role in global carbon dynamics. Parkland agroforestry has the potential to sequester carbon at an average rate of 0.4 tC ha$^{-1}$ yr$^{-1}$, which, if expanded to its maximum potential extent, would correspond to an additional carbon stock of approximately 558 TgC compared to treeless croplands. However, land surface models (LSM) used in global climate modeling struggle to represent carbon dynamics in these ecosystems due to the inadequate representation of deep-roots tapping groundwater during dry periods, key environmental control for many agroforestry systems such as the widespread parklands based on the phreatophytic species *Faidherbia albida*. This study explores the sensitivity of *Faidherbia albida* parklands to tree density and water availability (rainfall and soil water content in the capillary fringe of the groundwater table) using a new configuration of the ORCHIDEE LSM. To this aim, the ORCHIDEE LSM was modified to simulate the growth of *Faidherbia albida* by simulating its inverted phenology based on forced temporal series of soil water content of soil layers between 4 m and 5 m and water saturation below 5 m and by adjusting the photosynthesis and carbon allocation parameters for *Faidherbia albida* and associated crops. The model was evaluated against independent eddy covariance and meteorological data from the Niakhar agroforestry site in Senegal. Simulation outputs were analyzed in terms of leaf area index (LAI), gross primary productivity (GPP), latent heat (LE), sensible heat (H) and net radiation (Rn). The model simulated tree GPP of 4.08 ± 0.21 tC ha$^{-1}$ yr$^{-1}$ compared to observed GPP of 5.06 ± 0.49 tC ha$^{-1}$ yr$^{-1}$. For croplands, the model produced GPP of 7.97 ± 0.89 tC ha$^{-1}$ yr$^{-1}$ compared to observed values of 7.78 ± 1.75 tC ha$^{-1}$ yr$^{-1}$. Simulations revealed that tree density positively influenced annual carbon uptake but reduced crop harvest at highest tree densities, indicating a trade-off between carbon sequestration and crop yield. Sensitivity analyses showed that interannual variability in soil water content in the capillary fringe of the

groundwater table and rainfall influenced differently crop, tree and ecosystem carbon and energy fluxes. Despite its strengths, the model exhibited limited responsiveness of tree productivity to soil water content variability in the capillary fringe of the groundwater table, highlighting the need for enhanced representation of water uptake by tree roots in the model. These findings emphasize the importance of accurately modeling both surface soil water and groundwater dynamics and phenology to predict the responses of semi-arid agroforestry systems to climate variability. This study enhances our understanding of carbon and energy flux partitioning in complex, water-stressed and groundwater dependent agroforestry systems.

Keywords: Sahelian agroforestry, *Faidherbia albida*, groundwater-dependent ecosystems, Land Surface Model.

## 1 Introduction

Although rainfall has long been recognized as the primary factor influencing vegetation distribution in the West African Sahel (Ndehedehe et al., 2019; Seghieri et al., 2009), analyses of 12-year time series of correlations between normalized difference vegetation index and water availability (Ndehedehe et al., 2016) have shown that terrestrial water storage serves as an additional driver of vegetation dynamics (Ndehedehe et al., 2019). Many Sahelian plants rely on deep root systems that access groundwater at depths of up to 40 meters (Antunes et al., 2018; Do et al., 2005), providing an important water source in arid landscapes (Huber et al., 2011). The reliance on both rainfall and groundwater is essential for vegetation resilience and contributes to the observed increase in vegetation cover in nearly 24 % of the Western and Central Sahel since the 1980s, a process now commonly referred to as "Sahel regreening" (Dardel et al., 2014; Fensholt et al., 2013; Herrmann et al., 2005). While regreening is largely driven by rainfall, human-induced changes in land use also play an important role. Agricultural practices, such as land clearing and the adoption of specific cropping systems, have altered soil properties, reduced transpiration, and increased surface runoff, which, in turn, has enhanced groundwater recharge (Charreau and Fauck, 1970; Scanlon et al., 2005, 2006). Groundwater levels in some areas of the Sahel have risen, even during periods of declining rainfall, such as the severe droughts of the 1970s and 1980s (Descroix et al., 2009; Favreau et al., 2009), highlighting the resilience of the Sahel region's hydrological systems (Leduc et al., 2001). Projections under future climate scenarios, however, suggest a decrease in West African rainfall, particularly in the western portion of it, with increasing severity at higher global warming levels (Dosio et al., 2021; Kumi and Abiodun, 2018) potentially challenging this resilience. Such projected decrease in rainfall could indeed lead to a decline in groundwater recharge and storage (Kotchoni et al., 2019; Toure et al., 2016), threatening the sustainability of both vegetation and water resources (Glanville et al., 2023; Yin et al., 2015).

*Faidherbia albida* parklands are agroforestry systems commonly found in the Sahelian landscape. These agroforestry systems are widespread due to the natural distribution of *Faidherbia albida* trees across a broad geographic area from Senegal to Ethiopia, including parts of Southern Africa (Barnes and Fagg, 2003; Maslin et al., 2003; Vandenbeldt et al., 1992; Wickens, 1969). *Faidherbia albida* parklands are a specific type of agroforestry system relying mainly on groundwater dynamics thanks to the phreatophyte behavior of *Faidherbia albida* trees (Barnes and Fagg, 2003; Roupsard et al., 1999). The trees' taproots

can extend down to 20 meters to access groundwater and extract nutrients from deeper soil layers, which it cycles back to the surface soil layers (Barnes and Fagg, 2003; Sileshi, 2016). At the plot scale, the annual water use of *Faidherbia albida* trees was estimated to represent less than 10% of the rainfall input (Roupsard et al., 1999; Sarr et al., 2023). This modest total use was shown to rely strongly on groundwater by stable isotope tracing (Roupsard et al., 1999). In environments where the groundwater is relatively shallow, *Faidherbia albida* was reported to have a bimodal root distribution, with a high density of fine roots in the shallow soil (30-60 cm), allowing the capture of surface nutrients, and an increase in root density near the groundwater, which allows for the extraction of groundwater from the capillary fringe (Gning et al., 2023; Roupsard et al., 1999; Siegwart et al., 2023). *Faidherbia albida* is also characterized by a peculiar reverse phenology, as its leaves typically develop during the dry season but shed during the rainy season. A strong benefit of the reverse phenology for farms is that trees can be pruned throughout the dry season for feeding cattle, and leaves and pods provide valuable forage for livestock during the dry season when other feed sources are scarce (Barnes and Fagg, 2003). This reverse phenology also reduces competition for water and light between trees and crops during the wet season and an increase in crop yield in the vicinity of trees was reported (Roupsard et al., 2020).

Land Surface Models (LSMs) describe biophysical processes within the soil-plant-atmosphere continuum at scales ranging from plots to the globe (Boucher et al., 2020; Lawrence et al., 2019). At the global scale, LSMs serve as the land surface components of Earth System Models, which are used by organizations such as the Intergovernmental Panel on Climate Change for climate projections (Meinshausen et al., 2024). Substantial uncertainties in simulating dryland ecosystems by current LSMs have been documented (Fawcett et al., 2022). These uncertainties underscore the need for model refinements and caution when interpreting LSM estimates of carbon dynamics in semi-arid regions such as the Sahel. In particular, LSMs often fail to adequately represent deep-rooted plants that access groundwater during dry periods (Bastos et al., 2022; MacBean et al., 2021).

The ORCHIDEE model is the land surface component of the Institut Pierre Simon Laplace — Earth System Model (Boucher et al., 2020; Krinner et al., 2005). ORCHIDEE is a global process-based terrestrial biosphere model that calculates the fluxes of carbon, nitrogen, water, and energy between the surface and the atmosphere (Krinner et al., 2005). ORCHIDEE can be coupled with a global climate model to allow the analysis of vegetation feedback and land use on the climate, or be used as a standalone model as in this study, forced by climate data, to assess the impact of climate on ecosystems, among other applications (Boucher et al., 2020).

Considering the beneficial effects of *Faidherbia albida* and their importance for Sahel as well as the uncertainty in the response of semi-arid ecosystems to future global warming, this study raises the question of the sensitivity of *Faidherbia albida* parklands to tree density and water (rain and water in the capillary fringe of the groundwater) availability. To address this question, the LSM ORCHIDEE (revision 7949) was specifically configured for *Faidherbia albida* to assess how tree density and water availability influence carbon and energy fluxes in dryland ecosystems. As a first approach on mature *Faidherbia*

*albida* trees, this development of ORCHIDEE assumed a permanent phreatophyte type relying only on the groundwater for water uptake and as a driver for phenology.

The objectives of this study are: (1) to evaluate the skills of the newly configured version of ORCHIDEE by comparing its simulations with independent data on the *Faidherbia albida* ecosystem, (2) to assess the sensitivity of tree and crop productivity, as well as energy and carbon fluxes, to variations in tree density and water availability.

## 2 Materials and methods

### 2.1 Study area

The study was conducted in the actively farmed agroforestry parkland of Niakhar, in the agricultural zone classified as the "groundnut basin" of Senegal, Western Africa (Fatick region, 135 km east of Dakar) (Fig. 1). The climate of Niakhar is Sudano-Sahelian, with a wet season ranging from June to October and a dry season ranging from November to May. Niakhar is an area of dynamic agro-silvo-pastoral fields dominated by the multifunctional tree *Faidherbia albida*. The main crops are pearl millet (*Pennisetum glaucum* (L.) R. Br., var. Souna) and groundnut (*Arachis hypogaea* L., var. 55–437), conducted in annual rotation. *Faidherbia albida* trees are spread across the landscape at a density of 13 trees ha$^{-1}$ in the study area just around the eddy covariance tower, equivalent to a canopy cover of 15 % as calculated from the average crown area of 116 m² per tree. Based on surveys of 3000 trees in Niakhar (Lalou R., Montes N., pers. comm.), and dendrochronology analysis (Dougabka D., pers. comm.), the average age of the *Faidherbia albida* trees in this site was estimated at 55 years, the average tree diameter and height were 0.67 m and 12.8 m, respectively.

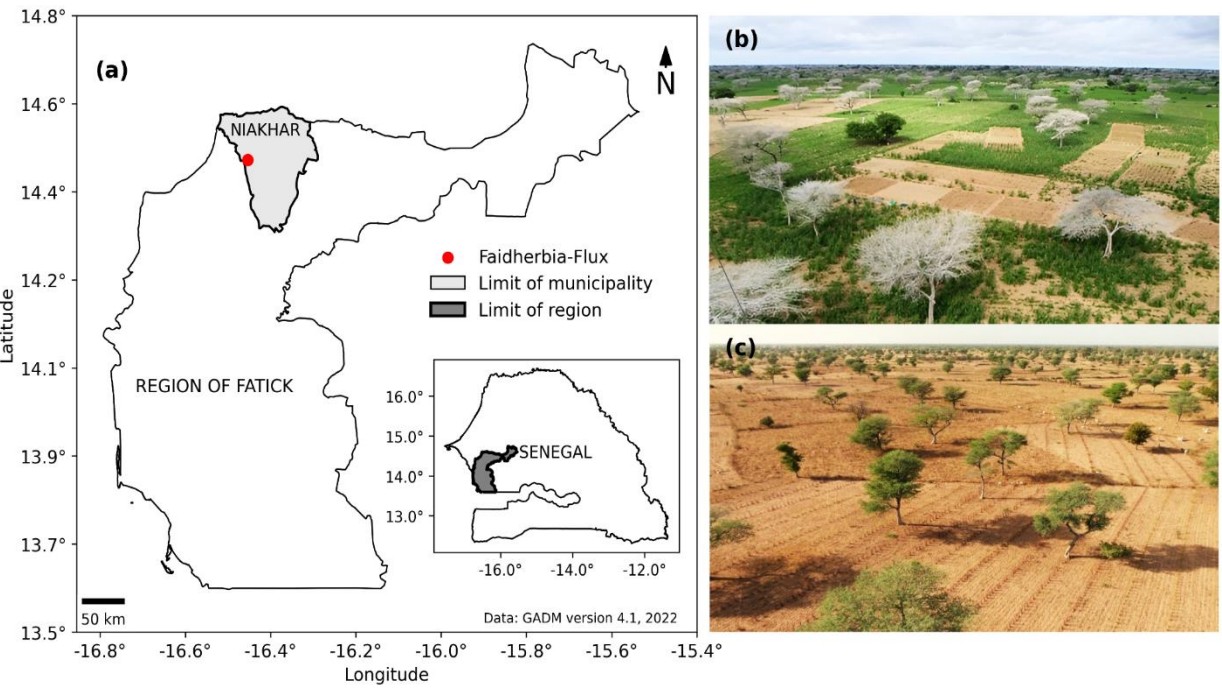

**Figure 1: (a) Location of the Faidherbia-Flux observatory (Niakhar, Senegal). Pictures of the agro-silvo-pastoral ecosystem dominated by the reverse phenology *Faidherbia albida* trees in Niakhar during the (b) wet (trees are defoliated, crops are growing) and (c) dry seasons (trees are in leaf, the soil is bare). Photos: Olivier Roupsard**

## 2.2 Faidherbia-Flux measurement setup

The data used in this study originate from the experimental site "Faidherbia-Flux", located at N: 14°29′44.916″; W: 16°27′12.851″ (Roupsard et al., 2020). The Faidherbia-flux database holds half-hourly flux estimates from eddy covariance measurements and the associated weather and ancillary variables.

### 2.2.1 Eddy covariance flux, climate variables and leaf area index

Half-hourly data of the "Faidherbia-Flux" site from 2018 to 2023 were extracted from the database. The eddy-covariance system (Li-COR SMARTFLUX including a Gill MasterPro 3D sonic anemometer and a LI-7500 RS open path $CO_2$ and $H_2O$ gas analyzer) was mounted at 20 m high on a 30 m high mast and monitored the ecosystem exchange of sensible heat (H), latent heat (LE), and $CO_2$ above the agroforestry system including trees, crops, and soil. Climate variables measured at 20 m high include air temperature and relative humidity (also measured at 2 m high), wind speed, atmospheric pressure, and rainfall by making use of the CS215 Campbell, the WindSonic 4 Campbell, the SMARTFLUX, and the Texas Electronics model TE525MM instruments, respectively. Net radiation was measured at 20 m by a nRlite net radiometer (Kipp & Zonen).

The eddy-covariance raw data were acquired at a frequency of 20 Hz using SMARTFLUX and post-processed using the advanced mode of the EddyPro v7.0 software by making use of the SMARTFLUX '.ghg' files. The following options were used in EddyPro: (1) raw data processing (automatic angle-of-attack correction for wind components (Gill sonic anemometers), double axis rotation for tilt correction, block average detrend method for turbulent fluctuations, covariance maximization with default for time-lag compensation); (2) compensation for density fluctuations (WPL terms, Webb et al., (1980)); (3) quality checks — flagging policy (Foken et al., 2005); (4) statistical tests for raw data screening following(Vickers and Mahrt, 1997); (5) estimation of random uncertainty due to sampling error (Finkelstein and Sims, 2001); (6) spectra and cospectra calculation and corrections (low-frequency range according to Moncrieff et al., (2005) and high frequency range according to Moncrieff et al., (1997)). Footprints were computed according to Kormann and Meixner, (2001), using the FREddyPro R package (Xenakis, 2016). Gap-filling and partitioning of $CO_2$ data were performed using the ReddyProc R package (Wutzler et al., 2018), selecting the daytime partitioning model of Lasslop et al., (2010). The partitioning of GPP between tree and crop takes advantage of the reverse phenology of *Faidherbia albida*: trees are completely defoliated during the rainy season, when crops are growing and are leafy during the dry season, when crops are absent. A simple temporal separation is therefore sufficient to distinguish tree and crop contributions.

Tree leaf area density was estimated by optical indirect measurements every ten days at 4 azimuths under 13 trees and above (in a nearby clearing) at the plot scale using the Li-Cor LAI2200, at dawn to have 100 % diffuse light and further analyzed following the 'isolated tree' protocol (Li-Cor, 2012; Taugourdeau et al., 2014). Tree leaf area index (LAI) was computed by multiplying the crown volume by leaf area density and dividing by the projected crown area. Tree LAI was filtered for anomalous data which occur when below and above canopy measurements do not perfectly represent the same sky area and gap-filled using visual estimation of the proportion of leaves in the crowns, monitored every ten days by one single operator.

Crop LAI dynamics were derived at the semi-hourly time-step from normalized difference vegetation index measurements using a proxy detection of normalized difference vegetation index sensor placed on the tall mast and directed to the crops (Pontailler et al., 2003), with data filtering according to Soudani et al., (2012) and an annual calibration of normalized difference vegetation index with the actual crop LAI from crop harvest in subplots, according to Diongue et al., (2022) and Roupsard et al., (2020).

### 2.2.2 Soil measurements

The dominant soil around Niakhar contains 88.1 ± 4.9 % sand, 6.1 ± 0.6 % silt, and 5.8 ± 1.6 % clay and is poor in organic matter (Diongue et al., 2022; Roupsard et al., 2020). The soil depth is about 8 m limited by the Eocene sandstone bedrock. Soil humidity at nine depths (15, 30, 50, 75, 100, 125, 150, 175, and 200 cm) were recorded with automated time domain reflectometers (CS615, Campbell Scientific, UK) (Diongue et al., 2022). Groundwater fluctuations between 5 and 6 m were recorded with piezometers and revealed a brackish water table at a depth of approximately 6 m (Diongue et al., 2023). The soil

water content in the capillary fringe of the groundwater (SWCC) was measured with a time domain reflectometer probe inserted at 4.8 m deep in a well.

### 2.2.3 Photosynthetic parameters (Vcmax and Jmax)

The rubisco activity (Vcmax) and the electron transport rate (Jmax) were measured from A_Ci and A_PAR curves every two weeks for 3 days in a row, in the same tree on sunlit leaves of the top canopy from January to April (Vandewalle, 2024).

**2.3 ORCHIDEE configuration for *Faidherbia albida* ecosystems**

This study makes use of ORCHIDEE revision 7949 which is referred to as the standard configuration of the model in this study. We adjusted the code of revision 7949 to the *Faidherbia albida* agroforestry systems by modifying the modules for soil, root, and phenology, and by adjusting the photosynthesis and carbon allocation and assimilation parameters (Fig. 2). The modified configuration (Table S1) is referred to as the "Faidherbia configuration".


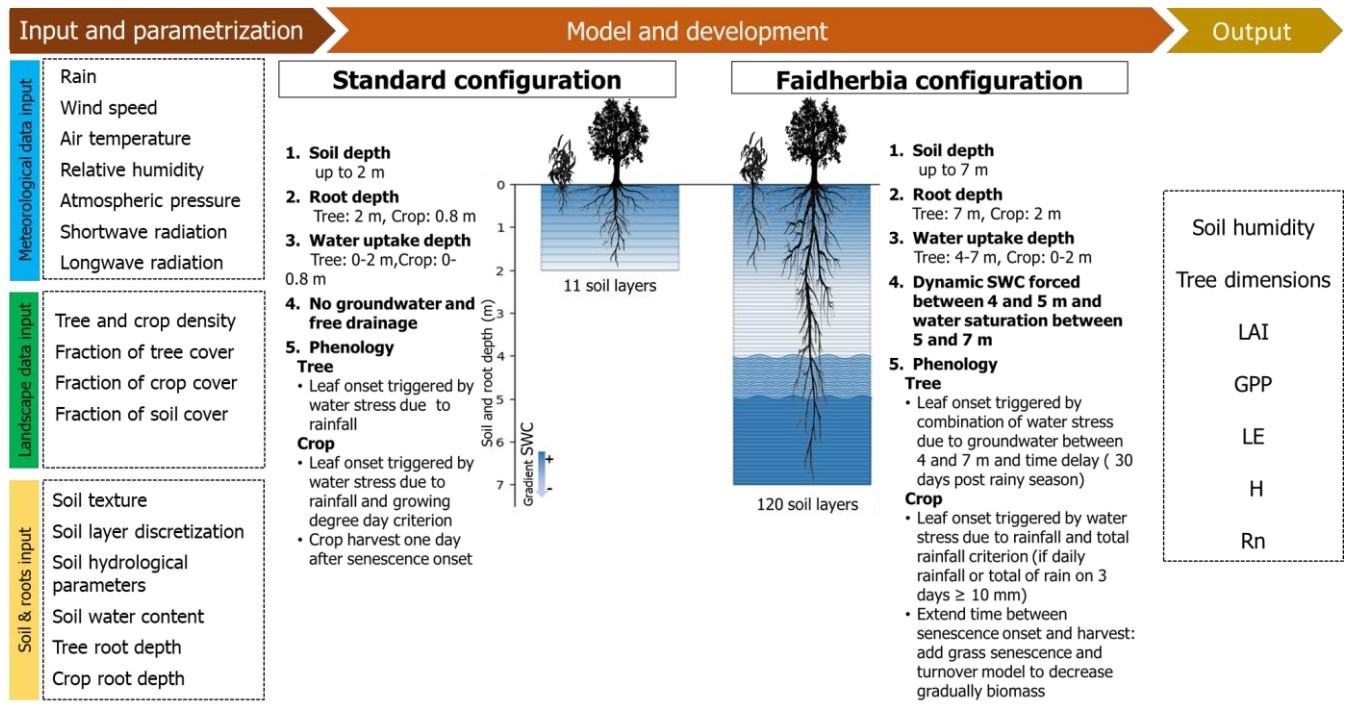

**Figure 2: Schematic diagram of the ORCHIDEE standard model and Faidherbia configurations showing the input data required, the process modifications and a non-exhaustive list of the model output variables used in this study. SWC: soil water content; LAI: leaf area index; GPP: gross primary productivity; LE: latent heat flux; H: sensible heat flux; Rn: net radiation.**


### 2.3.1 Vegetation description

Vegetated areas in the model are defined by up to 14 plant functional types (PFTs) in addition to bare soil. A PFT is characterized by a classification scheme that takes into account morphology (tree or grass), leaf morphology (needleleaf or broadleaf), phenology (evergreen, summer-green, or rain-green), photosynthetic pathway (C3 and C4), and climatic zones (boreal, temperate and tropical) (Krinner et al., 2005; Poulter et al., 2015). Each PFT is assigned a fraction of the pixel area (the sum of all PFT fractions being 1).

The soil water column is divided into three hydrological tiles with homogeneous soil hydrological properties (Boucher et al., 2020) , one for bare soil, one for short vegetation, i.e. croplands and grasslands, and one for tall vegetation, i.e. trees. Within each soil tile all PFTs share the same water, inducing competition, whereas there is no interaction between the water consumption of vegetation in different soil tiles. The energy budget is calculated for each PFT independently and then averaged according to the area fraction of each PFT into a pixel-level energy budget. There is therefore no competition for light between

different PFTs.

For this study, three PFTs were used: bare soil, tropical deciduous summer-green trees, and C4 crop with areal cover fractions of 0.1, 0.15, and 0.75 respectively to simulate the observed tree density.

### 2.3.2 Soil and root configuration

The soil representation in the standard configuration of ORCHIDEE is based on a multi-layer physical description of the soil.

By default, the soil depth is set at 2 m (Fig. 2) and the soil is discretized into 11 layers of increasing thickness following a geometric series with a ratio of two (Campoy et al., 2013; de Rosnay et al., 2002). In the Faidherbia configuration, the soil depth was increased to 7 m (Fig. 2), and the soil was discretized into 120 layers following Ducharne et al., (2018) to ensure numerical schemes' stability. A finer resolution was used in the upper soil layers with a geometrical increase of soil layers from surface to 0.15 m (ratio: 1.8) to capture the strong gradient of soil moisture and temperature near the surface. From 0.15

m to 6.5 m, a constant layer thickness of 0.065 m is applied to simplify the model where gradients are less pronounced. Between 6.5 m and 7 m, the geometric progression resumes to ensure numerical stability as strong soil moisture gradients are again modelled as a result of free drainage at the soil bottom.

Soil texture in the standard configuration, defined by the fractions of sandy loam, loam, and clay loam, is homogeneous in the pixel and controls the soil parameters, i.e., field capacity and wilting point, Van Genuchten parameters (Boucher et al., 2020),

which are assumed constant through time. In the Faidherbia configuration, soil texture and soil hydrological parameters were taken from Diongue et al., (2022) and Roupsard et al., (2020) as summarized in Table S1.

In the standard configuration, the bottom soil layers allow free drainage which prevents simulating groundwater dynamics. In the Faidherbia configuration, groundwater was simulated by forcing a dynamic water content of soil layers between 4 m and 5 m and imposing water saturation below 5 m, following Campoy et al., (2013). Groundwater forcing was exclusively implemented for the hydrological tile dedicated to tree PFTs.

In ORCHIDEE, the structural root profile defines the root biomass distribution in the soil and hence depends on the carbon allocation. In turn, the functional root profile defines which soil layers plants can draw water from. As only the functional profile was modified in this configuration, root profile hereafter refers to this functional profile, which is defined following Eq. (1):

$$Rf(i) = \begin{cases} 0, & \text{if } i = 1 \\ 0, & \text{if } \sum_{i=2}^{n} \max\left(0, W_i - W_{p,i}\right) \leq 0 \\ \frac{\max(0, W_i - W_{p,i})}{\sum_{i=2}^{n} \max(0, W_i - W_{p,i})}, & \text{if } i > 1 \text{ and } \sum_{i=2}^{n} \max\left(0, W_i - W_{p,i}\right) > 0 \end{cases} \qquad (1)$$

where i is the index for each soil layer, Rf represents the root profile at position i (unitless) and ranges between 0 and 1, n is the total of soil layers, $W_i$ represents the soil moisture at position i (kg m$^{-2}$) and $W_{p,i}$ represents the wilting point moisture at position i (kg m$^{-2}$).

In the standard model configuration, maximum crop root depth is 0.8 m and maximum tree root depth is 2 m (Fig. 2). In the Faidherbia configuration, crop and tree root depths were increased to 2 m and 7 m respectively to match field observations and the root profile was adjusted to be partially consistent with recent observations of Siegwart et al., (2023). It was assumed that crops use water from soil layers up to 2 m in depth, whereas tree roots take up water from depths below 4 m (Fig. 2). Tree water uptake from superficial roots was ignored, an assumption supported by their observed low contribution to total tree water use (Roupsard et al., 1999) and is further explained in the discussion section (4.1 and 4.2). Thus from Eq. (1), the adjusted root profile for *Faidherbia albida* is calculated following Eq. (2):

$$Rf(i) = \begin{cases} 0, & \text{if } i < i_{4m} \\ 0, & \text{if } i \geq i_{4m} \text{ and } \sum_{i=i_{4m}}^{n} \max\left(0, W_i - W_{p,i}\right) \leq 0 \\ \frac{\max(0, W_i - W_{p,i})}{\sum_{i=i_{4m}}^{n} \max(0, W_i - W_{p,i})}, & \text{if } i \geq i_{4m} \text{ and } \sum_{i=i_{4m}}^{n} \max\left(0, W_i - W_{p,i}\right) > 0 \end{cases} \qquad (2)$$

where i is the index for each soil layer, $i_{4m}$ represents the index of the soil layer at 4 m, Rf represents the root profile at position i (unitless) and ranges between 0 and 1, n is the total of soil layers, $W_i$ represents the soil moisture at position i (kg m$^{-2}$) and $W_{p,i}$ represents the wilting point moisture at position i (kg m$^{-2}$). All soil and root depths, and soil water content parameters used in this study are summarized in Table S1.

### 2.3.3 Phenology and photosynthesis parameters

In the standard configuration, leaf onset for tropical trees is driven by soil moisture constraints on growth (Eamus et al., 2013; Murphy and Lugo, 1986) and is therefore modelled as a function of water stress as calculated over the whole soil column

(Botta et al., 2000). The detailed calculation of water stress is provided in the Supplement (Text S1). Leaf onset can be triggered by three conditions: (1) either when a given time has passed since the minimum soil moisture for plant growth was recorded, and (2) the weekly moisture availability is higher than monthly moisture availability, or (3) when the monthly moisture availability exceeds a prescribed critical moisture availability. In the Faidherbia configuration, a time delay was added to these previous water stress conditions based on observations by Roupsard et al., (2022), who found that the first leaves of *Faidherbia albida* appeared on average 30 days after groundwater rise (Fig. 2).

In the standard model configuration, the phenology for the crop PFT is based on a combination of growing degree days (Chuine, 2000) and water stress (Botta et al., 2000). In the Faidherbia configuration, the same temperature and water stress conditions apply but a rainfall amount threshold was added, specifying that leaf onset can only occur if there is a minimum of 10 mm of rainfall over three days (Berg et al., 2010; Marteau et al., 2011; Ndiaye et al., 2024). This constraint ensures that leaf emergence is not triggered by very minor precipitation events.

In the standard configuration, crops are harvested abruptly just after the senescence stage starts, unlike trees, in which leaf biomass gradually dies during the senescence period. This abrupt crop harvest does not reflect the harvesting practices of farmers in the study region, because the most common practice is to pull or cut the panicle of the millet stalk just below where the grain is formed (Hedayetullah and Zaman, 2022). The stalks can then be harvested from 80 cm above the ground (Kajuna, 2021; Lericollais, 1999), leaving some leaves which regrow at the internodes and thus allowing photosynthetic processes to continue before complete stalk mortality. In the Faidherbia configuration, in order to represent this process, a parameter was defined as a delay between senescence start and harvest adding a senescence stage in the crop life cycle. The newly defined crop senescence stage reproduces senescence and turnover processes used in the standard model for grasslands (Fig. 2) where biomass die-off depends on the growing degree days and leaf lifespan consistent with millet photoperiodism (Adole et al., 2019; Sanon et al., 2014). This adjustment results in a slow biomass decay until harvest when all parts of the crops are removed.

**2.3.4 Carbon assimilation for trees**

In the standard configuration, ORCHIDEE simulates C3 plant leaf-level photosynthesis using the extended Farquhar et al., (1980) model from Yin and Struik, (2009). This model considers the influence of rubisco activity (Vcmax) and electron transport rate (Jmax) on $CO_2$ assimilation. Most model parameters follow Yin and Struik, (2009) except Vcmax and Jmax, which are constrained by nitrogen use efficiency and acclimate to growth temperature (Kattge and Knorr, 2007; Medlyn et al., 2002). In the standard configuration, Vcmax is calculated from leaf nitrogen content at the top of the canopy and nitrogen use efficiency (NUE) of Vcmax which is prescribed for each PFT. In contrast, Jmax is calculated as a linear function of Vcmax and growth temperature (Kattge and Knorr, 2007). In the Faidherbia configuration, site observations of Vcmax and Jmax were used to parameterize NUE and Jmax (section 2.2.3).

## 2.4 Climate input data sets

ORCHIDEE requires climate input data of air temperature (°C), precipitation (mm), relative humidity (%), atmospheric pressure (hPa), wind speed (m s$^{-1}$), shortwave radiation (W m$^{-2}$), and long-wave radiation (W m$^{-2}$). Although ORCHIDEE is run at the half-hourly time step, the data could be at the half-hourly to 6-hourly time-step. If the frequency of the data is less than half-hourly, ORCHIDEE interpolates the climate forcing in between two data points. Three climate data sources were used as inputs in this study:

(1) When no in-situ climate data were available (i.e. before 2018), CRUJRA v2.2.2 produced by the Climatic Research Unit (CRU) (Harris et al., 2020) and Japanese reanalysis (JRA) data (Kobayashi et al., 2015) were used. CRUJRA provides a 6-hourly, 0.5-degree global reconstruction from 1901 to 2020.

(2) In-situ observations were used for 2018 to 2023. In-situ measurements started on 22 February 2018 and thus contain a gap from January 1$^{st}$ to February 21$^{st}$ of 2018. To address this issue, the meteorological data were gapfilled following

Vuichard and Papale, (2015). Short gaps are interpolated, but for longer gaps, the gap-filling procedure relies on the ERA Interim climate reanalysis. ERA Interim is the latest atmospheric reanalysis product provided by the European Centre for Medium-Range Weather Forecasts (ECMWF) (Dee et al., 2011). It offers data from 1989 to present on a regular spatial grid of 0.7 degrees and a temporal resolution of 3 hours. For years 2018 to 2023, meteorological variables were available from Faidherbia-flux observatory with measurements with a half-hourly time step (Text S2

and Fig. S2 in the Supplement).

(3) To eliminate artifacts that would occur when switching from CRUJRA to in-situ observations, a statistical linear correction was applied to CRUJRA data for the period 1967-2019. This correction aligned the mean values of the common years in CRUJRA (2018-2020) with those observed in the data from 2018-2020, following the approach proposed by Vuichard and Papale, (2015) for ERA Interim. This method corrects biases in CRUJRA data by

establishing a linear relationship with an observed dataset for most meteorological variables. The regression coefficients for global radiation and wind speed fields were calculated with an intercept forced to zero to prevent negative radiation values and an excessively flat regression slope for wind speed. An alternative method, based on ratios was used for precipitation data due to timing inaccuracies, i.e., the moment of precipitation may differ between the CRUJRA and the in-situ observations. The ratio of the sum of monthly rainfall in the observed data to the sum

of monthly rainfall in CRUJRA was calculated. The de-biased precipitation field was then obtained by multiplying CRUJRA rain data by this ratio.

## 2.5 Simulation protocol

The simulation protocol consists of the following eight runs: (1) spin-up, (2) CO$_2$ transient, (3) climate transient, (4) clear-cut, (5) experiment, (6) calibration, (7) evaluation, and (8) sensitivity runs. Runs (1) to (4) include only the tree PFT, while runs

(5) to (8) include both tree and crop PFTs. The details of each run are provided below:

(1) The spin-up run aims at initializing the soil and litter carbon and nitrogen masses and is run until an equilibrium of the system's carbon, nitrogen and water pools is achieved. For this, the spin-up consists of a 500-year semi-analytical spinup with the entire pixel assumed to be covered by tropical deciduous summergreen trees, weather cycling over the 1901 to 1910 time series of meteorological variables (CRUJRA) and atmospheric $CO_2$ concentration fixed at the 1860 level and thus equal to 286.42 ppm.

(2) A first transient run aims at including the effect of rising $CO_2$ on the carbon, nitrogen and water fluxes and the carbon pools. For this, a 40 years transient simulation is run with weather still cycling over the 1901-1910 meteorological forcing and the atmospheric $CO_2$ progressively increasing from 286.45 ppm to the 1901 level of 296.57 ppm.

(3) A second transient run aims at matching climate and $CO_2$ historical levels. It consists of 60 years run with 1901-1961 CRUJRA climate forcing and linear increase in $CO_2$ from 296.80 ppm to 317.09 ppm.

(4) A clear-cut run aims at constraining the trees' ages to match the observed average tree age of 55 years in 2017. It consists of a one-year run for 1962 with a forest clear-cut. Living biomass is removed and transferred to the litter pools before a new stand can grow.

(5) A 55-year experimental simulation (1963-2017) aims at getting the ecosystem to the 2018 conditions through simulated growth. It consists of 55 years run with tree and crop cover forced with 1963-2017 CRUJRA climate forcing data and historical atmospheric $CO_2$ concentration increasing from 318.40 ppm to 404.71 ppm.

(6) A calibration simulation was run to adjust model parameters. For this, a 3-year simulation for years 2018-2020 was run (section 2.6) with climate forcing from observed in situ climate data (section 2.4). After obtaining the calibrated parameters from step 6, steps 1 to 6 were then repeated with those parameters.

(7) The evaluation simulation aiming at evaluating the robustness of model calibration with previously unseen data relied on a 2021-2023 simulation with observed in situ climate data.

(8) Sensitivity runs aim at evaluating the effect of tree density or water (rain and water in the capillary fringe of the groundwater) availability on ecosystem functioning (section 2.8). They consist of 3- or 6-years long simulations using observed in situ climate data and scenarios of tree density and water availability.

## 2.6 Model calibration

First, model allometry was calibrated by adjusting the parameters related to tree allometric properties, such as wood density, form factor for cylinder volume reduction, and height factors (Table S1)) until trees dimensions approximately matched observed diameters and height presented in section 2.1.

In a second step, parameters determining the start and end of the growing season in the model phenology, including minimum time since season start, minimum time elapsed, leaf longevity, and length of leaf senescence (Table S1), were adjusted through visual comparison of simulated and observed LAI on three years for crops (2018-2020) and two years for trees (2019-2020).

For crops and trees subsequently, starting from model default values, parameters were increased or decreased until dry and wet season dynamics for LAI were within less than 7 days of observed data.

In a third step, crop and tree carbon and energy fluxes were calibrated sequentially focusing on LAI, GPP, sensible (LE) and latent (H) heat. For this, three years for crops (2018-2020) and two years for trees (2019-2020) of observed and simulated data were compared using root mean square error (RMSE). In this third step, parameters were incrementally varied until a local minimum was reached for RMSE. The leaf-to-sapwood area ratio was adjusted for LAI until local minimization of RMSE and an acceptable maximum LAI value was reached (Table S1). In this ORCHIDEE configuration, maximum LAI is determined prognostically by the carbon allocation scheme rather than being set as a prescribed parameter. Consequently, we calibrated the leaf-to-sapwood area ratio to serve as a structural constraint to govern the simulated maximum LAI. For GPP, parameters associated with photosynthesis and carbon allocation, such as light absorption efficiency, electron transport, and Rubisco kinetics, were adjusted until local minimization of RMSE and an acceptable maximum GPP value was achieved (Table S1). Due to interaction between LAI and GPP variables, calibration of GPP called for a second adjustment loop of the leaf-to-sapwood area ratio. As a result of the large uncertainties in LAI compared to GPP measurements, priority was given to GPP seasonal maximum calibration. For the energy flux (LE and H), evapotranspiration and vegetation structure parameters were fine-tuned to ensure consistency with observed LE and H patterns aiming for minimized RMSE and a realistic maximum of LE and H.

**2.7 Model evaluation performance**

To ensure the obtained calibrated parameters were not too dependent on the specific years and on the variables used for calibration, three years of observations (2021-2023) were used to evaluate the model fitness outside the calibration conditions. Simulations were compared against daily observations of LAI, GPP, LE, H and net radiation as well as annual total GPP with RMSE and $r^2$ as indicators of model fitness. Statistical analysis in the calibration and evaluation were performed using Python v3.9 (Python Software Foundation – available at http://www.python.org).

**2.8 Sensitivity analysis**

Two sensitivity analyses were performed in order to determine:

(1) the sensitivity of ecosystem GPP, above-ground harvested crop biomass and energy fluxes (LE and H) to tree density. For this, four planting densities were applied to the model: 0 trees ha$^{-1}$, 7 trees ha$^{-1}$, 13 trees ha$^{-1}$ (observed density in the plot), and 26 trees ha$^{-1}$. For the sensitivity analysis to tree density, all areas not occupied by *Faidherbia albida* trees were assumed to be cultivated with crops. Since ORCHIDEE uses a self-thinning relationship that links stand density to tree diameter, the PFTs proportions rather than the actual densities were adjusted in the pixel to obtain the expected tree densities over the whole pixel. Assuming an average crown area of 116 m² for 13 trees ha$^{-1}$, tree densities of 0, 7, 13, and 26 trees ha$^{-1}$ respectively corresponded to 0 %, 7.5 %, 15 %, and 30 % tree PFT fraction in the pixel (Table S2).

Statistical analyses were performed using the "stats" package (v4.41) within R software (R Core Team, 2024). The suitability of data for parametric tests was assessed by checking for variance homogeneity with the Bartlett test and evaluating residuals for normality using the Shapiro-Wilk test. One-way ANOVAs were applied when these conditions were satisfied. When assumptions for parametric tests were not met, the Kruskal-Wallis test, a non-parametric alternative, was employed. Post-hoc pairwise comparisons for ecosystem GPP and above-ground harvested crop biomass were conducted using Tukey's honestly significant difference test.

(2) the sensitivity of GPP and energy fluxes (LE and H) to year-to-year fluctuations in rainfall and soil water content in the capillary fringe of the groundwater table (SWCC). For this, the average yearly cycle of precipitation and groundwater dynamics were respectively computed to be used as multi-year time series without any interannual variability. For SWCC, the average cycle was reconstructed from key observed dates and amplitudes of the groundwater fluctuation (1 Jan: 0.26 m³ m⁻³; 8 Jul: 0.15 m³ m⁻³; 10 Sep: 0.15 m³ m⁻³; 24 Oct: 0.31 m³ m⁻³; 31 Dec: 0.26 m³ m⁻³), and linear interpolation was applied between these points to produce a smoothed climatological cycle (Fig. S3). These are called the "average" scenarios (see Text S3 and Fig. S3 for details on the calculation in the Supplement). The sensitivity analysis consisted of four climate and SWCC combinations: i) simulation with average rain and average SWCC ($R_{avg}SWCC_{avg}$), ii) simulation with average rain and variable SWCC ($R_{avg}SWCC_{var}$), iii) simulation with variable rain and average SWCC ($R_{var}SWCC_{avg}$) and iv) simulation with variable rain and variable SWCC ($R_{var}SWCC_{var}$). In all simulations, the other climate forcing variables were taken from CRUJRA climate forcing data.

The sensitivity is quantified as the anomaly calculated as the daily difference of all scenarios with respect to $R_{avg}SWCC_{avg}$ considered as the reference scenario. The anomalies in SWCC, GPP, LE and H for Faidherbia trees (scenario $R_{avg}SWCC_{var}$), were analyzed over the dry season, that is, from the beginning of October in year n to the end of June in year n+1. In contrast the anomalies in rainfall, GPP, LE and H for crops (scenario $R_{avg}SWCC_{var}$) were analysed over the rainy season, that is, from the beginning of July to the end of September of the same year. Given the variability of both rainfall and SWCC in the scenario $R_{var}SWCC_{var}$, anomalies in GPP, LE, and H at ecosystem level were assessed over annual periods covering both dry and rainy seasons.

## 3 Results

### 3.1 Model evaluation

#### 3.1.1 Phenology (LAI)

For trees, the 'reverse phenology' of *Faidherbia albida* with its growing season in the dry season was reproduced by the model with an RMSE of 0.15. Simulated and observed peaks in LAI were aligned with the dry growing season and the wet dormant season of this tree species (Fig. 3e). Over the evaluation period, a maximum measured tree LAI of 0.81 was recorded, while maximum simulated LAI values reached 0.73. However, during 2 out of 3 evaluation years, the maximum tree LAI was

overestimated by the model by 11 % and 13 % in 2020-2021 and 2022-2023 respectively (Fig. 3e). The interannual variability
of the simulated tree LAI was lower in the simulations than in the observations with a standard deviation of the maximum LAI
of 0.002 in simulations compared to 0.09 in observations, and a standard deviation of the maximum LAI day relative to the
emergence day is 1 day in simulations and 17 days in observations over the evaluation period (Fig. 3e).

Regarding crops, the model simulated a maximum daily LAI of 0.90 compared to the observed maximum daily crop LAI of
1.20 with an RMSE of 0.26 over the evaluation period (Fig. 3f). With a single parameter set being used irrespective of the crop
species, the model goodness-of-fit did not appear to depend on the actual crop sown. The day of maximum LAI was simulated
with a delay of +17, -7, -10 days in 2021 to 2023 respectively (Fig. 3f). For crops, similar to trees, the model underestimated
the interannual variability of LAI over the evaluation period. Specifically, the maximum crop LAI was underestimated by 28
%, 24 %, and 22 % in 2021, 2022, and 2023, respectively (Fig. 3f).

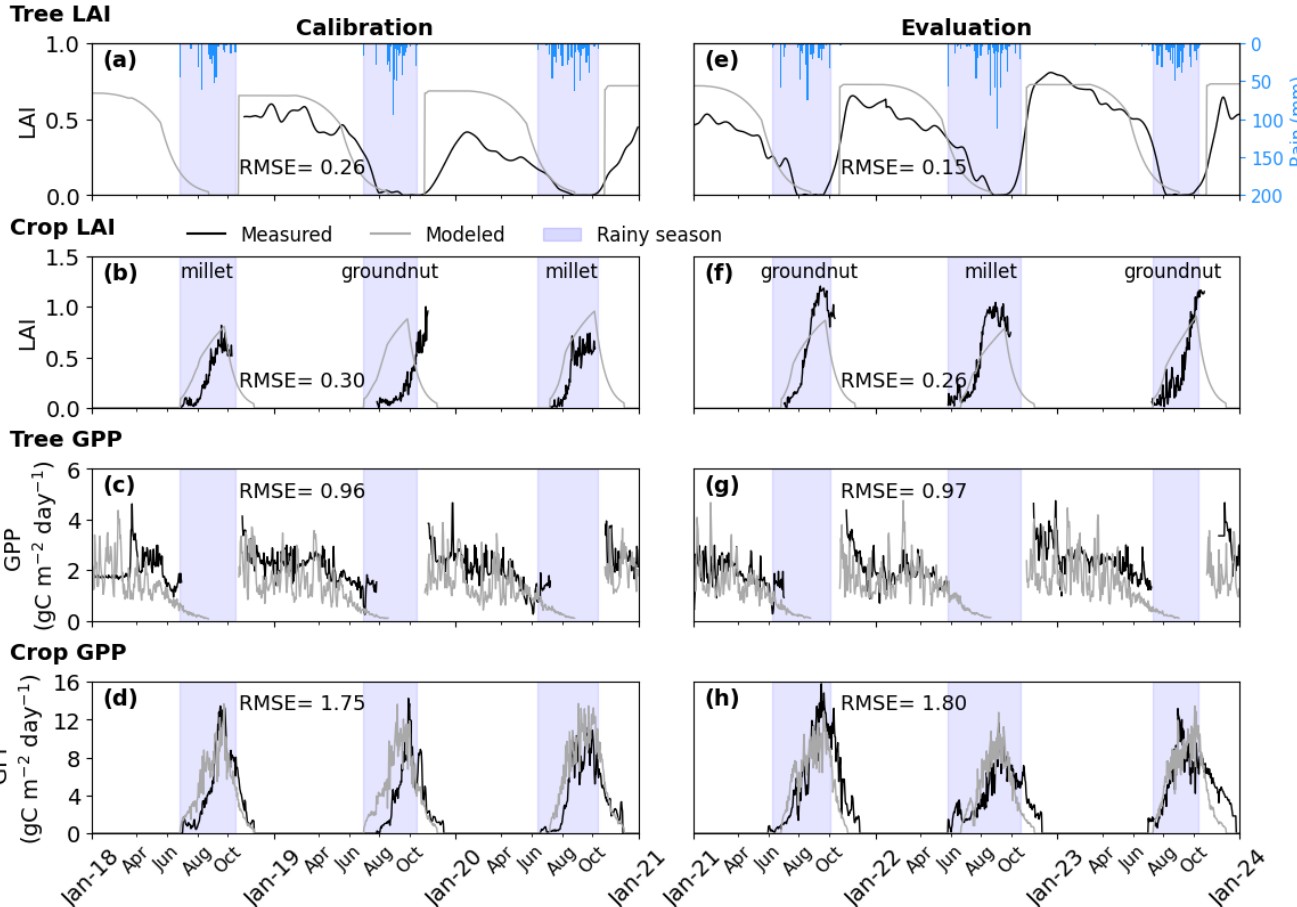

**Figure 3: Measured and modeled time series at the daily time-step for calibration and evaluation periods for (a, c, e, g) tree and (b,**
**d, f, h) crop, showing leaf area index (LAI) and gross primary productivity (GPP), respectively. The crops, millet and groundnut,**

are cultivated in an annual rotation. Shaded blue areas indicate the rainy season. Annual rainfall from 2018 to 2023 was 454.43 mm, 513.05 mm, 599.34 mm, 482 mm, 821.62 mm, and 537.83 mm, respectively.

### 3.1.2 Carbon sequestration (GPP)

The observed daily tree GPP reached a peak value of 4.74 gC m$^{-2}$ day$^{-1}$, closely matched by the model's maximum prediction
of 4.75 gC m$^{-2}$ day$^{-1}$ (Fig. 3g). During the evaluation period, the r$^2$ between simulation and observation of the daily tree GPP was 0.16 (n=666) and RMSE of 0.97 gC m$^{-2}$ day$^{-1}$. However, the model consistently underestimated the annual GPP of trees, with errors ranging from 2 % to 22 % throughout the evaluation period (Table 1).

For crops, the observed daily GPP peaked at 15.76 gC m$^{-2}$ day$^{-1}$, while the model's maximum prediction was lower, at 13.41 gC m$^{-2}$ day$^{-1}$ (Fig. 3h). The RMSE for daily crop GPP was 1.80 gC m$^{-2}$ day$^{-1}$, and the r² between simulated and observed
values was 0.76 in the evaluation (n = 1095; Fig. 3d). The model underestimated the peak daily crop GPP in 2021 (Fig. 3h). On an annual scale, the crop GPP was underestimated by 9 % to 19 % between 2021 and 2023 (Table 1).

Day-to-day variability in EC-derived GPP was lower than that of simulated GPP. This variability was pronounced for trees (Figs. 3c and 3g) and present but weaker for crops (Figs. 3d and 3h). The coefficient of variation over the whole period for the simulated GPP is 0.6 for trees and 1.6 for crops compared to 0.3 and 1.7 for observed tree and crop GPP.

For the whole ecosystem, the total annual GPP was underestimated by 18 %, 6 %, and 16 % in 2021, 2022, and 2023, respectively (Table 1). Tree contributions to the total annual ecosystem GPP varied between 32 % and 37 % in observations and between 33 % and 37 % in simulations during the 2021–2023 period.

**Table 1: Observed and simulated annual GPP for trees and crops over the study period.**

| | | Calibration | | | Evaluation | | | |
|---|---|---|---|---|---|---|---|---|
| Annual GPP | | 2018 | 2019 | 2020 | 2021 | 2022 | 2023 | mean ± standard deviation |
| Tree | Observed (tC ha$^{-1}$ yr$^{-1}$) | 5.22 | 5.51 | 5.12 | 4.76 | 4.23 | 5.51 | 5.06 ± 0.49 |
| | Modeled (tC ha$^{-1}$ yr$^{-1}$) | 4.31 | 3.77 | 3.93 | 4.03 | 4.14 | 4.31 | 4.08 ± 0.21 |
| | Relative bias (%) | -17 | -32 | -23 | -15 | -2 | -22 | — |
| Crop | Observed (tC ha$^{-1}$ yr$^{-1}$) | 6.81 | 5.41 | 7.07 | 10.02 | 7.81 | 9.58 | 7.78 ± 1.75 |
| | Modeled (tC ha$^{-1}$ yr$^{-1}$) | 7.26 | 7.47 | 9.49 | 8.11 | 7.12 | 8.35 | 7.97 ± 0.89 |
| | Relative bias (%) | 7 | 38 | 34 | -19 | -9 | -13 | — |
| Ecosystem scale | Observed (tC ha$^{-1}$ yr$^{-1}$) | 12.03 | 10.92 | 12.19 | 14.79 | 12.04 | 15.10 | 12.84 ± 1.69 |
| | Modeled (tC ha$^{-1}$ yr$^{-1}$) | 11.57 | 11.24 | 13.42 | 12.15 | 11.26 | 12.66 | 12.05 ± 0.87 |
| | Relative bias (%) | -4 | 3 | 10 | -18 | -6 | -16 | — |

### 3.1.3 Energy budget (LE, H and Rn)

The daily simulated latent heat flux reached a maximum of 11.56 MJ m$^{-2}$ day$^{-1}$ compared to the daily observed maximum of 12.66 MJ m$^{-2}$ day$^{-1}$ in the evaluation (Fig. 4d). The model tended to overestimate low daily latent heat flux values by the end of the dry seasons between 2021 and 2023. The evaluation yielded an RMSE of 2.12 MJ m$^{-2}$ day$^{-1}$ and a correlation between simulated and observed daily latent heat flux of 0.49 (n=1095) (Figs. 4d and 4g).

The maximum simulated sensible heat flux was 11.42 MJ m$^{-2}$ day$^{-1}$, exceeding the observed maximum of 9.46 MJ m$^{-2}$ day$^{-1}$

between 2021 and 2023 (Fig. 4e). The evaluation indicated an r$^2$ of 0.67 and an RMSE of 2.40 MJ m$^{-2}$ day$^{-1}$ for daily sensible heat flux (n = 1095). However, the model consistently overestimated both low and high daily sensible heat flux values (Figs. 4e and 4h).

For net radiation, the simulated daily maximum was 16.08 MJ m$^{-2}$ day$^{-1}$, higher than the observed daily maximum of 14.42 MJ m$^{-2}$ day$^{-1}$ in the evaluation (Fig. 4f). Between 2021 and 2023, the model consistently overestimated daily net radiation

during the dry season. Over the evaluation period, the RMSE was 2.31 MJ m$^{-2}$ day$^{-1}$, and the model explained 72 % of the variability in observed daily net radiation (Figs. 4f and 4i).

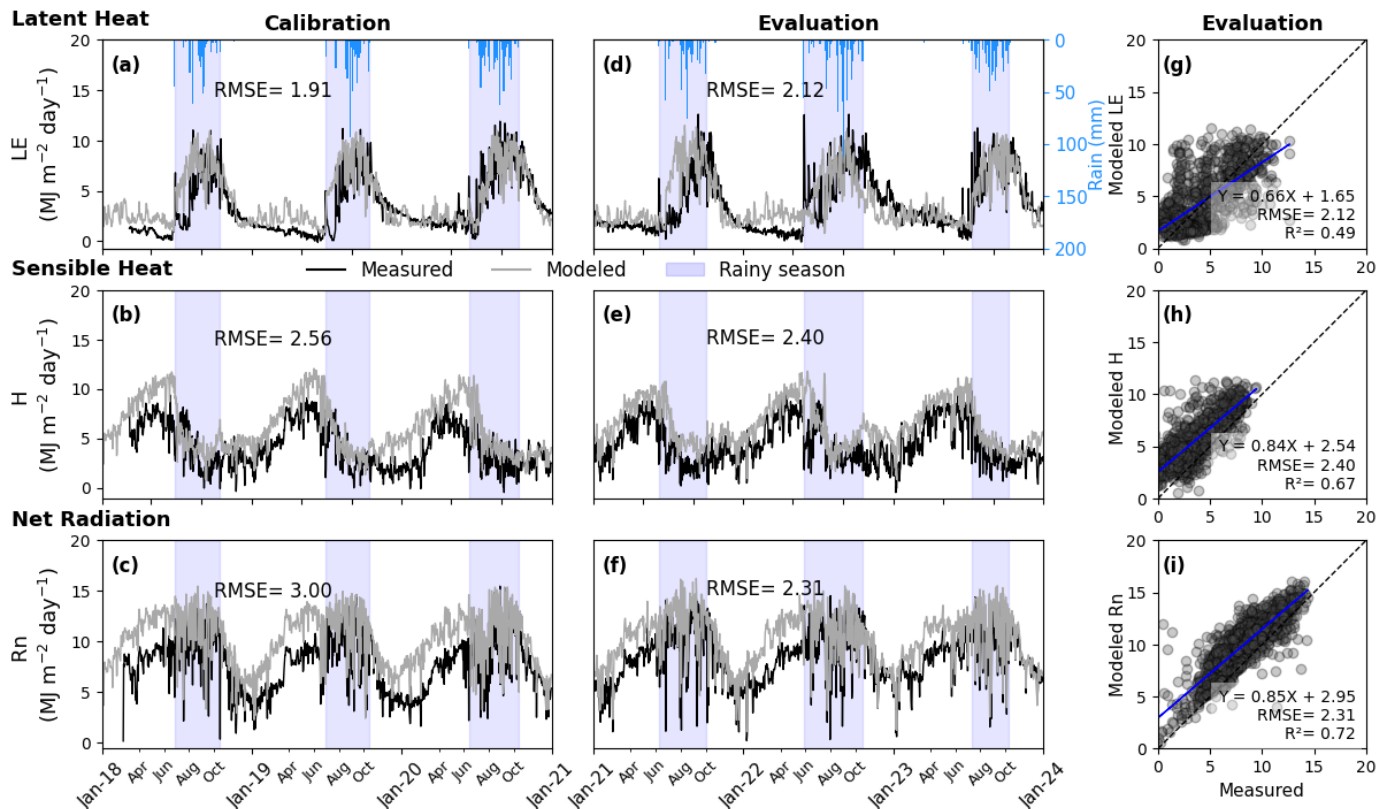

**Figure 4: Daily time series (left) and scatter plots (right) comparing modeled and measured (a, d, g) latent heat flux (LE), (b, e, h) sensible heat flux (H), and (c, f, i) net radiation (Rn). The left panels show calibration and evaluation periods, while the right panels focus on evaluation scatter plots. Shaded blue areas indicate the rainy season. Annual rainfall from 2018 to 2023 was 454.43 mm, 513.05 mm, 599.34 mm, 482 mm, 821.62 mm, and 537.83 mm, respectively. The dashed line represents the 1:1 line, while the solid blue line depicts the linear regression line.**

## 3.2 Ecosystem sensitivity to tree density

Varying tree density (0, 7, 13, and 26 trees per hectare) significantly influenced annual GPP during the dry season (Kruskal-Wallis, n = 24, p < 0.001; Table 2). Annual GPP increased during the dry season (when Faidherbia trees grow) with tree density, ranging from 0 tC ha$^{-1}$ yr$^{-1}$ at 0 trees ha$^{-1}$ to 8.64 ± 0.84 tC ha$^{-1}$ yr$^{-1}$ at 26 trees ha$^{-1}$ (Table 2). In contrast, during the rainy season, when crops grow and Faidherbia trees are dormant, tree density had no significant effect on annual GPP for densities of 0 to 7 trees ha$^{-1}$. However, from the densities of 13 to 26 trees ha$^{-1}$, annual GPP decreased from 7.96 ± 0.89 tC ha$^{-1}$ yr$^{-1}$ to 6.33 ± 0.86 tC ha$^{-1}$ yr$^{-1}$ (Table 2). Across the entire ecosystem, regardless of season, annual GPP was significantly higher at 26 trees ha$^{-1}$ (14.97 ± 1.32 tC ha$^{-1}$ yr$^{-1}$) compared to 0 trees ha$^{-1}$ (9.91 ± 0.64 tC ha$^{-1}$ yr$^{-1}$) (ANOVA, n = 24, p < 0.001; Fig. 5a and Table 2).

Above-ground harvested crop biomass decreased with increasing tree density. Biomass was not significantly different between 0 and 7 trees ha$^{-1}$ (3.33 ± 0.26 tC ha$^{-1}$ yr$^{-1}$ and 3.1 ± 0.28 tC ha$^{-1}$ yr$^{-1}$, respectively), nor between 7 and 13 trees ha$^{-1}$ (2.77 ±

0.33 tC ha$^{-1}$ yr$^{-1}$). However, biomass at 13 trees ha$^{-1}$ was significantly lower than at 0 trees ha$^{-1}$, and a further significant reduction was observed at 26 trees ha$^{-1}$ (2.08 ± 0.34 tC ha$^{-1}$ yr$^{-1}$) (Fig. 5b and Table 2).

During the dry season, tree density significantly affected ecosystem LE (ANOVA, n = 24, p < 0.001), with higher densities resulting in higher LE values, ranging from 0.12 ± 0.07 GJ m$^{-2}$ yr$^{-1}$ at 0 trees ha$^{-1}$ to 0.85 ± 0.12 GJ m$^{-2}$ yr$^{-1}$ at 26 trees ha$^{-1}$ (Table 2). In the rainy season, LE did not significantly differ between 0 and 7 trees ha$^{-1}$ (0.72 ± 0.07 and 0.7 ± 0.07 GJ m$^{-2}$
yr$^{-1}$, respectively) but showed a slight reduction at 13 trees ha$^{-1}$ (0.67 ± 0.06 GJ m$^{-2}$ yr$^{-1}$) and a significant decrease at 26 trees ha$^{-1}$ (0.58 ± 0.07 GJ m$^{-2}$ yr$^{-1}$, p < 0.01, Table 2). No significant effect of tree density on ecosystem H was observed during the dry season at densities of 0 to 13 trees ha$^{-1}$, with values ranging from 1.73 ± 0.11 to 1.61 ± 0.1 GJ m$^{-2}$ yr$^{-1}$. However, at 26 trees ha$^{-1}$, H decreased significantly to 1.51 ± 0.12 GJ m$^{-2}$ yr$^{-1}$ (p < 0.01, Table 2). In the rainy season, H did not significantly differ between 0 and 7 trees ha$^{-1}$ (0.33 ± 0.03 and 0.35 ± 0.04 GJ m$^{-2}$ yr$^{-1}$, respectively) but increased slightly at 13 trees ha$^{-1}$
(0.38 ± 0.03 GJ m$^{-2}$ yr$^{-1}$) and significantly at 26 trees ha$^{-1}$ (0.43 ± 0.06 GJ m$^{-2}$ yr$^{-1}$, p < 0.01, Table 2).

**Table 2: Annual statistical summary of gross primary productivity (GPP in tC ha$^{-1}$ yr$^{-1}$), energy fluxes (LE and H in GJ m$^{-2}$ yr$^{-1}$) and crop harvest (tC ha$^{-1}$ yr$^{-1}$) across tree densities (0, 7, 13, and 26 trees). n = 24 (6 years × 4 tree density conditions). Values followed by different letters are significantly different.**

| Season | Variable | mean ± standard deviation | | | | Test | p-value |
|---|---|---|---|---|---|---|---|
| | | 0 trees | 7 trees | 13 trees | 26 trees | | |
| Dry | Tree GPP | 0 ± 0d | 2.16 ± 0.21c | 4.08 ± 0.21b | 8.64 ± 0.84a | Kruskal-Wallis | <0.001 |
| | LE | 0.12 ± 0.07d | 0.33 ± 0.08c | 0.62 ± 0.1b | 0.85 ± 0.12a | ANOVA | <0.001 |
| | H | 1.73 ± 0.11a | 1.74 ± 0.11a | 1.61 ± 0.1ab | 1.51 ± 0.12b | ANOVA | <0.01 |
| Rainy | Crop GPP | 9.91 ± 0.64a | 9.28 ± 0.71a | 7.96 ± 0.89b | 6.33 ± 0.86c | ANOVA | <0.001 |
| | Crop harvest | 3.33 ± 0.26a | 3.1 ± 0.28ab | 2.77 ± 0.33b | 2.08 ± 0.34c | ANOVA | <0.001 |
| | LE | 0.72 ± 0.07a | 0.7 ± 0.07a | 0.67 ± 0.06ab | 0.58 ± 0.07b | ANOVA | <0.01 |
| | H | 0.33 ± 0.03b | 0.35 ± 0.04b | 0.38 ± 0.03ab | 0.43 ± 0.06a | Kruskal-Wallis | <0.01 |
| - | Ecosystem GPP | 9.91 ± 0.64c | 11.44 ± 0.74b | 12.05 ± 0.87b | 14.97 ± 1.32a | ANOVA | <0.001 |

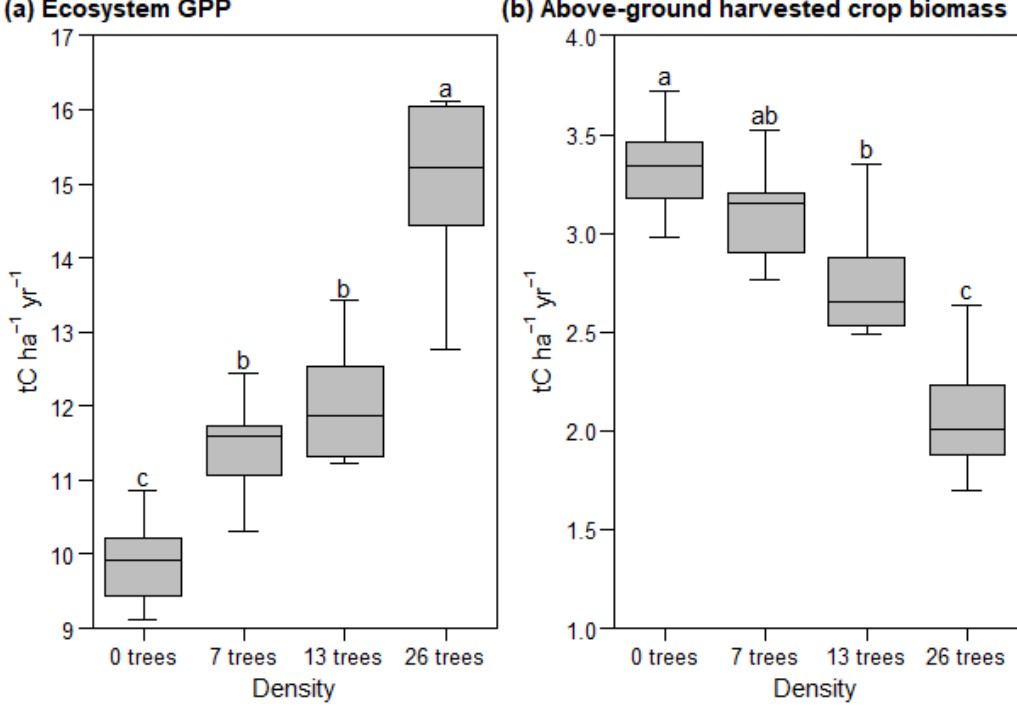

**Figure 5: Effect of the Faidherbia tree density on (a) modeled ecosystem gross primary productivity (GPP) and (b) modeled above-ground harvested crop biomass over six years. The GPP and above-ground harvested crop biomass values represent the annual ecosystem totals for each tree density scenario, averaged over six years. Denotation with different letters indicates significant differences, as revealed by a Tukey (HSD) Post hoc-Test ($\alpha = 0.05$).**

### 3.3 Ecosystem sensitivity to interannual water variability

### 3.3.1 Water variability

Compared to the reference scenario with no variability in the yearly cycle of groundwater availability, $SWCC_{var}$ scenarios mimic: (i) an average deficit of 19 % from October 2019 to June 2020 and (ii) an average excess of 27 % from October 2020 to June 2021 (Figs. 6a and S5a). Compared to the reference scenario with no variability in rainfall, the $R_{var}$ scenarios capture distinct rainfall patterns over the study years: (i) an annual rainfall deficit of 63 mm (12 %) in 2019, primarily driven by an early-season deficit of approximately 66 mm in July, (ii) an annual rainfall excess of 53 mm (10 %) in 2020, notably due to an early-season surplus of about 84 mm in July, and (iii) an annual rainfall deficit of 146 mm (28 %) in 2021, characterized by alternating periods of excess, such as 50 mm in July, and deficits, including 134 mm in September (late-season) (Figs. 7a and S6a).

### 3.3.2 GPP

As expected, variations in groundwater dynamics only affected tree GPP as the crop roots are too shallow to rely on groundwater (Fig. 6). For trees, accounting for groundwater variability resulted in a similar pattern in both years with a decrease in tree GPP compared to the reference at the start of the Faidherbia growing season and an increase in GPP towards the end of the season corresponding to a shift in the growing period of the Faidherbia trees (Figs. 6b and S5b). Since the early-season GPP deficit is larger than the late-season excess, these two response stages combined result in an overall decrease in tree GPP of less than 10 gC m$^{-2}$ yr$^{-1}$, corresponding to approximately 2 % in both 2019–2020 and 2020–2021, during the dry seasons for the $R_{avg}SWCC_{var}$ and $R_{var}SWCC_{var}$ scenarios (Figs. 6b and 8b).

As expected, rainfall variability primarily influenced crop GPP (Figs. 7a-b), given that tree phenology is shifted to the dry season and tree roots in the model exclusively access groundwater resources. For crops, the GPP response to rainfall variability differed depending on the timing of the anomalies. In 2019, where rainfall was below the reference and early-season deficits were predominant, crop GPP decreased by at least 20 gC m$^{-2}$ yr$^{-1}$ (less than 4 %) under both $R_{var}SWCC_{avg}$ and $R_{var}SWCC_{var}$ scenarios (Figs. 7b and 8d). Conversely, in 2020, the rainfall surplus, particularly in early-season, led to an increase in crop GPP, with gains of approximately 65 gC m$^{-2}$ yr$^{-1}$ (11 %) under the same scenarios. Despite a severe late-season rainfall deficit in 2021, crop GPP did not show a corresponding decline. Instead, early-season rainfall excess appeared to offset the deficit, resulting in a crop GPP increase of about 86 gC m$^{-2}$ yr$^{-1}$ (14 %) for both scenarios (Figs. 7c and 8c).

At the ecosystem level, in the $R_{var}SWCC_{var}$ scenario, where both rainfall and soil water content in the capillary fringe of the groundwater table varied (Figs. 8a-d), simulations indicated a decrease in ecosystem GPP of approximately 78 gC m$^{-2}$ yr$^{-1}$ (about 8 %) in 2019. In contrast, during 2020 and 2021, ecosystem GPP increased by 10 % to 14 %, corresponding to approximately 118 gC m$^{-2}$ yr$^{-1}$.

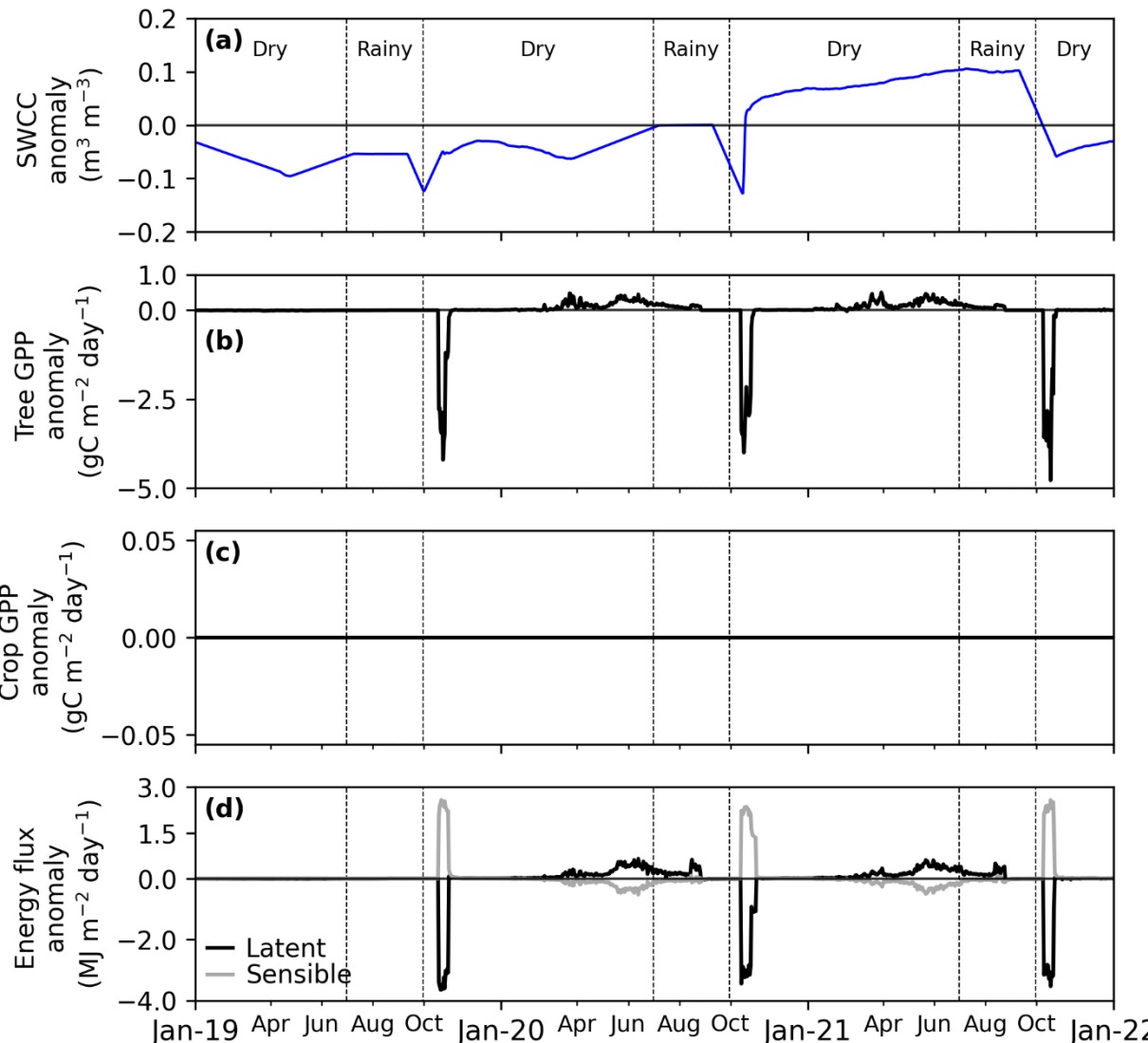

**Figure 6: Sensitivity study of the effect of daily anomalies in soil water content in the capillary fringe of the groundwater table (SWCC) with rainfall unmodified: (a) soil water content in the capillary fringe of the groundwater table (SWCC), (b) tree gross primary productivity (GPP), (c) crop GPP anomaly, (d) latent and sensible flux anomaly. RavgSWCCvar is a simulation with average rain and variable SWCC, thus affecting the tree and the ecosystem, but not the crop. The sensitivity is quantified as the anomaly of the RavgSWCCvar scenario with respect to RavgSWCCavg (simulation with average rain and average SWCC), considered as the reference scenario.**

505

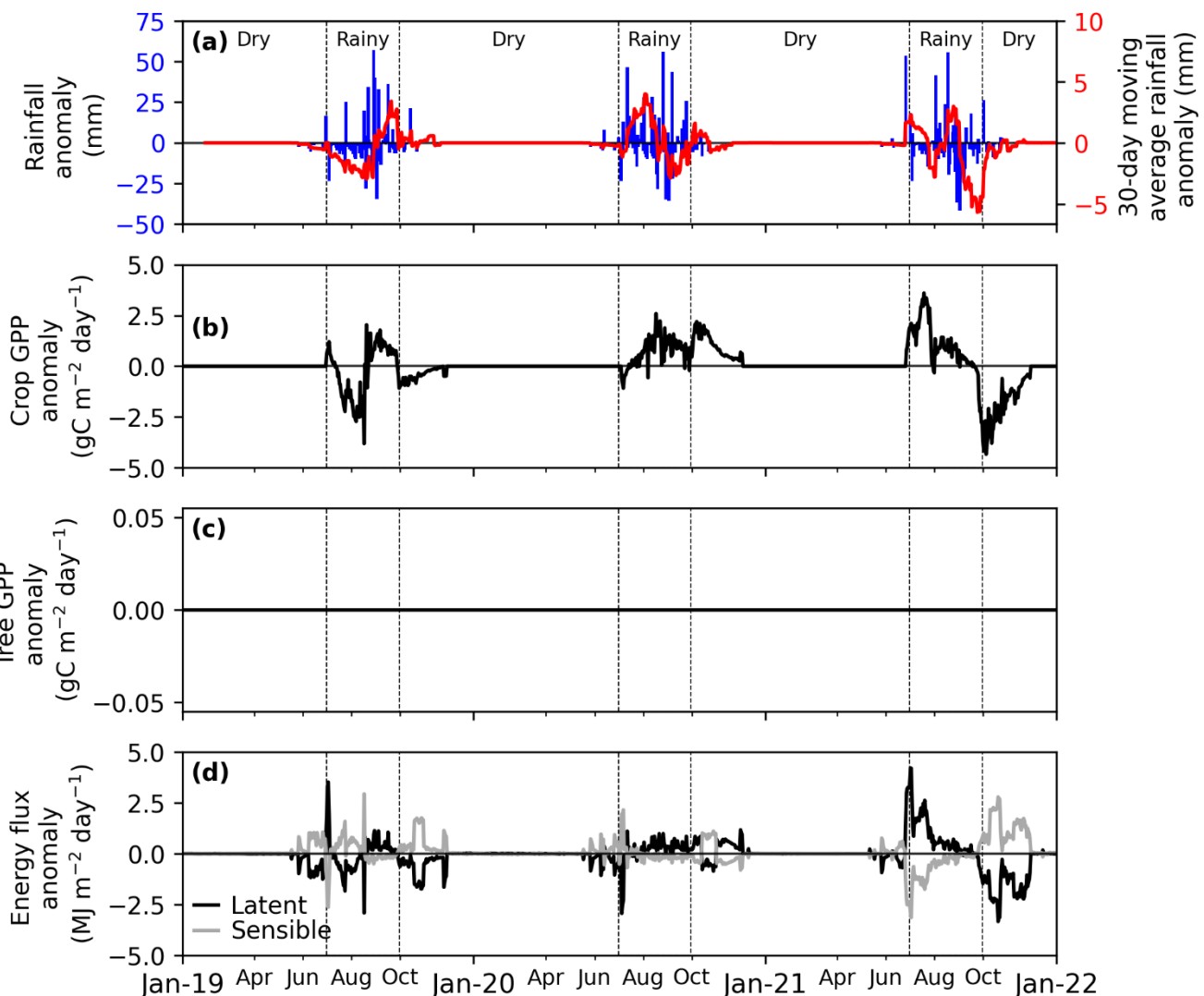

**Figure 7: Sensitivity study of the effect of daily anomalies in rainfall (SWCC unmodified) (a) rainfall anomalies: daily (in blue line) and 30-day moving average (in red line), (b) crop gross primary productivity (GPP) anomaly, (c) tree GPP anomaly, (d) latent and sensible flux anomalies. RvarSWCCavg is a simulation with variable rain and average SWCC, thus affecting the crop and the ecosystem, but not the tree. The sensitivity is quantified as the anomaly of the RvarSWCCavg scenario with respect to RavgSWCCavg (simulation with average rain and average SWCC), considered as the reference scenario.**

510

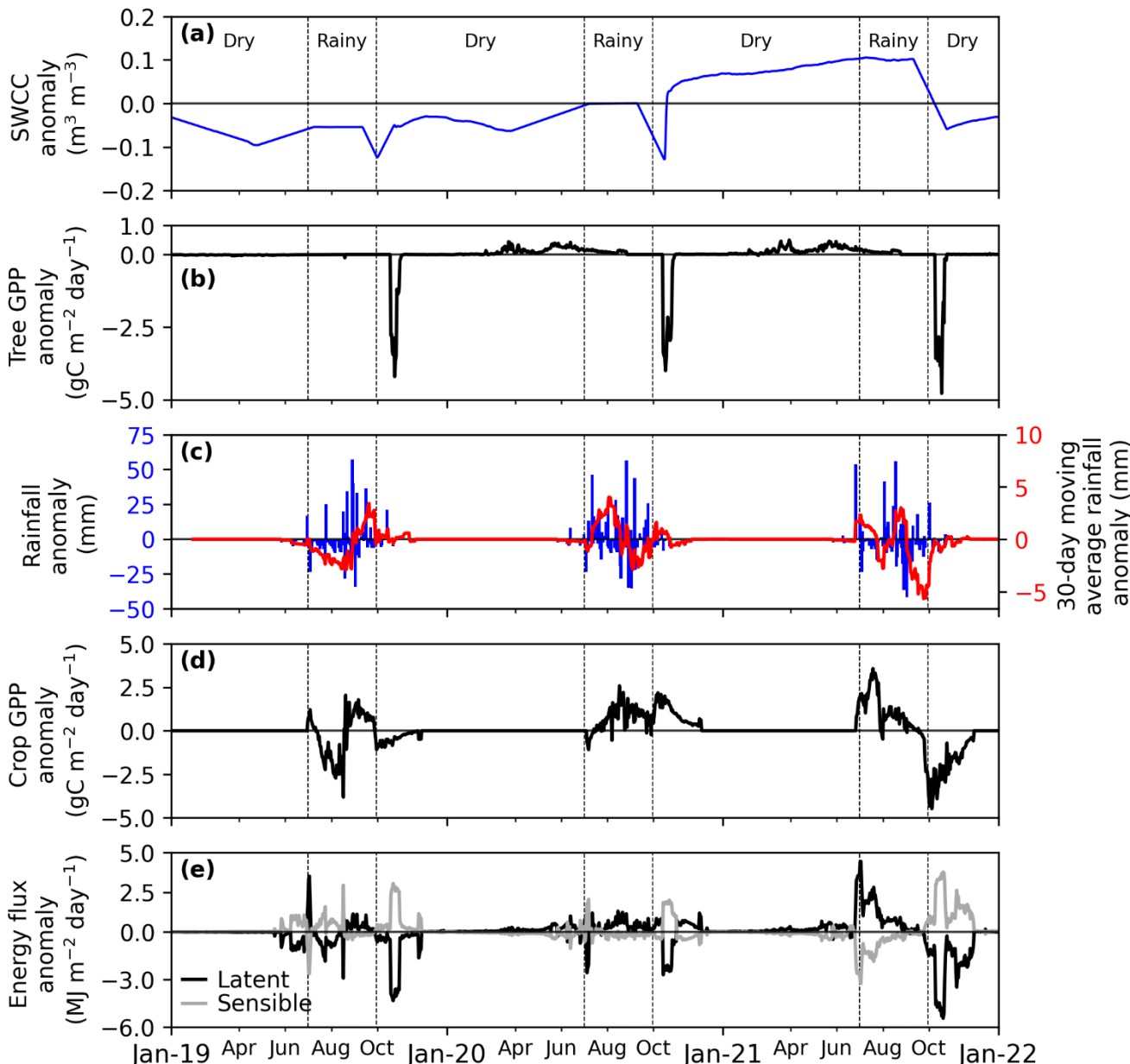

Figure 8: Sensitivity study of daily anomalies in rainfall and in SWCC (a) soil water content in the capillary fringe of the groundwater table (SWCC) anomaly, (b) tree gross primary productivity (GPP) anomaly, (c) rainfall anomaly: daily (in blue line) and 30-day moving average (in red line), (d) crop GPP anomaly, (e) latent and sensible flux. RvarSWCCvar is a simulation with variable rain and variable SWC, where crop, tree, and ecosystem are affected. The sensitivity is quantified as the anomaly of the RvarSWCCvar scenario with respect to RavgSWCCavg (simulation with average rain and average SWCC), considered as the reference scenario.

### 3.3.3 Energy fluxes (LE and H)

During the dry seasons, energy flux (LE and H) anomalies in the $R_{avg}SWCC_{var}$ scenario were strongly correlated with tree GPP anomalies (Figs. 6b-d). LE decreased by 8 MJ m$^{-2}$ yr$^{-1}$ (<1 %) during 2019–2020 and by 11 MJ m$^{-2}$ yr$^{-1}$ (<1 %) during the 2020–2021 dry season. In contrast, H increased from 5 MJ m$^{-2}$ yr$^{-1}$ to 11 MJ m$^{-2}$ yr$^{-1}$ over the same periods (Figs. 6d and S5d).

In the $R_{var}SWCC_{avg}$ scenario, where only rainfall varied annually, energy flux anomalies were driven by crop GPP anomalies (Figs. 7b-d). LE declined by approximately 2 MJ m$^{-2}$ yr$^{-1}$ (<1 %) in 2019 but increased from 14 MJ m$^{-2}$ yr$^{-1}$ (<3 %) in 2020 to 68 MJ m$^{-2}$ yr$^{-1}$ (10 %) in 2021. During the 2019 rainy season, H increased by approximately 9 MJ m$^{-2}$ yr$^{-1}$ (<3 %), whereas it declined by 2 MJ m$^{-2}$ yr$^{-1}$ (<1 %) in 2020 and by 48 MJ m$^{-2}$ yr$^{-1}$ (11 %) in 2021 (Figs. 7d and S6d).

In the $R_{var}SWCC_{var}$ scenario, energy flux anomalies were influenced by anomalies in tree GPP, crop GPP, and rainfall (Fig. 8). Consistent with ecosystem GPP anomalies, LE decreased by approximately 54 MJ m$^{-2}$ yr$^{-1}$ (5 %) in 2019 but increased from 60 MJ m$^{-2}$ yr$^{-1}$ (4 %) in 2020 to 116 MJ m$^{-2}$ yr$^{-1}$ (11 %) in 2021. In 2019, H increased by 41 MJ m$^{-2}$ yr$^{-1}$ (6 %), while it declined by 35 MJ m$^{-2}$ yr$^{-1}$ (<3 %) in 2020 and by 81 MJ m$^{-2}$ yr$^{-1}$ (5 %) in 2021 (Figs. 8e and S7e).

## 4 Discussion

### 4.1 Evaluation of the phreatophyte configuration in ORCHIDEE

A configuration of the ORCHIDEE r7949 model was developed to represent the phreatophytic behaviour of *Faidherbia albida* trees in Sahelian agroforestry systems. Model evaluation was conducted by comparing simulated LAI, carbon, and energy fluxes with site observations, a standard approach for assessing model performance (Collins et al., 2011; Li et al., 2016; Vuichard et al., 2019).

The model reproduced average LAI values reasonably well for both trees and crops, though interannual variability was not fully captured. For trees, simulated maximum LAI matched measured values but failed to reflect the observed year-to-year fluctuations (Figs. 3a and 3e). This limitation is primarily due to lack of realism in the allocation of carbon in this ORCHIDEE's configuration, which subsequently limits the interannual plasticity of leaf biomass. Additionally, simplified tree phenology (MacBean et al., 2015) represents a secondary contributing factor. Similarly, crop LAI dynamics compared well with simulations from the process-based model LandscapeDNDC (Rahimi et al., 2021), yet interannual variability remained underrepresented, primarily due to lack of realism in the allocation of carbon in the model, alongside simplified crop phenology and the use of a single crop PFT without accounting for species rotation.

Eddy-covariance-derived GPP is inferred from empirical models fitted to the net ecosystem exchange (NEE) rather than measured directly. Only NEE represents the full semi-hourly variability of the flux, whereas GPP and ecosystem respiration

(Reco) are estimated from moving-window fits of light- and temperature-response functions (Lasslop et al., 2010; Reichstein et al., 2012). This procedure inherently smooths short-term fluctuations, leading to a reduction of day-to-day variability in the partitioned components. Unlike EC-derived GPP, the simulated GPP shown stronger day-to-day variability, particularly during the dry season (Figs. 3c and 3g). This reflects the role of vapor pressure deficit (VPD) computed from the meteorological forcing (air temperature, specific humidity and surface pressure; Fig. S2) which limits assimilation via the stomatal scheme (Yin and Struik, 2009). Consequently, short-term changes in atmospheric dryness are transmitted directly to GPP (Fig. S4). The effect is stronger for trees than for crops, consistent with higher stomatal sensitivity in the tree parameterization.

The introduced configuration increased the process realism of the model by explicitly linking tree phenology to groundwater availability, in line with observations of *Faidherbia albida* water use (Roupsard et al., 1999; Sarr et al., 2023). At the same time, this came at the cost of reduced transferability. The use of exogenous groundwater forcing and parameter calibration based on three years of data at a single site limits the broader applicability of this setup. For example, the current approach is unlikely to capture crop dynamics in regions with different rainfall regimes, soil conditions, or crop management practices. Similarly, the assumption that trees rely exclusively on groundwater omits the possibility of mixed water sources, which may help explain the interannual variability of LAI observed at the site. The configuration improves representation of phreatophyte-driven phenology in ORCHIDEE at the case study site, but its wider applicability remains constrained by data availability and site-specific calibration. Extending the approach to other contexts will require the integration of a dynamic groundwater module coupled with surface water processes, as well as crop and species-specific parameterizations that account for local management and phenological diversity.

## 4.2 Increased tree density increases carbon sequestration and latent heat flux

ORCHIDEE's Faidherbia configuration was used to assess the impact of *Faidherbia albida* density scenarios on carbon and energy fluxes. Our study revealed a positive relationship between tree density and ecosystem GPP (Fig .5a). This relationship was a consequence of increasing tree canopy cover and LAI with trees sequestering more carbon per planted area than crops.

Energy flux analysis showed seasonal variations, with both latent heat and sensible heat responding to tree density, consistent with the simulated changes in tree GPP (Table 2). Higher tree densities lead to increased latent heat during the dry season, likely due to higher transpiration rates of trees (Wang et al., 2020) that overcame the reduction in latent heat in the rainy crop-growing season. This is consistent with the result of Sarr et al., (2023), showing a linear dependency between tree transpiration assessed by sapflow and LAI. The inverse relationship between sensible heat and tree density in the dry season suggested that more trees provide shade, reducing soil heat flux, thereby decreasing the Bowen ratio.

ORCHIDEE did not simulate grain yield, a critical indicator for farmers; however, the results showed that increasing tree density reduced the above-ground harvested crop biomass (Fig. 5b). In the model, this reduction results from the decrease in crop area, when tree density, hence tree area, increases with crop biomass and carbon stocks being reported per unit area of

the entire pixel. The simulated decline in crop biomass therefore reflects the competition for space. The trade-off between maintaining an optimal density of *Faidherbia albida* in agroforestry systems and achieving adequate above-ground harvested crop biomass and carbon stocks suggested an ideal range of 7 to 13 trees per hectare, where the difference remains not significant (Fig. 5). This range was consistent with traditional recommendations from the local Sereer community, which historically maintained approximately 7 trees per field (around 1 ha) to secure sufficient yields for a family's food needs (R. Diatte, pers. Comm.; Lericollais, 1972; Pélissier, 1966).

While many studies have highlighted the benefits of *Faidherbia albida* for both crops and the ecosystem (Diongue et al., 2023; Gning et al., 2023; Siegwart et al., 2022; Sileshi, 2016), recent observations suggest highly complex tree-crop interactions (Clermont-Dauphin et al., 2023). Investigation through model studies requires additional processes to be implemented in the model: (1) even though the *Faidherbia albida*'s root architecture, is partially implemented with a representation of a dense root network below the crop roots, the current model omits the shallow roots reported below the surface (Siegwart et al., 2023), and (2) the observed concentration of water and nutrients in the soil under the canopy of *Faidherbia albida* trees (Siegwart et al., 2022; Sileshi, 2016) creating a fertility 'island effect' that can be linked to aridity and plant traits (Eldridge et al., 2024). The improvement of soil organic matter resulting from higher surface water availability and nutrient accumulation could explain the 'island effect' in the close vicinity of the trees and the progressive reduction in millet yields with increasing distance from the tree (Diene et al., 2024; Roupsard et al., 2020).

Additionally, the feedback between tree groundwater consumption and groundwater recharge is currently omitted, hindering the possibility to draw conclusions on tree survival at high densities, especially in the context of the alternance of long drier and wetter periods in the Sahel. Considering these processes of water and nutrient redistribution would enable accounting for competition both among trees and between trees and crops for water and nutrients, thereby helping to determine an optimal tree density that maximizes carbon sequestration and surface cooling through latent heat flux while sustaining crop production.

## 4.3 Divergent GPP and energy fluxes responses of trees and crops to water availability

The similarity in the simulated response of tree GPP in both years with a water deficit at the start of the season but either positive or negative soil water content in the capillary fringe of the groundwater table anomaly during the growing season shows that GPP responds to the timing of the groundwater dynamics rather than total water availability during the growing season. This result appears counterintuitive, as interannual fluctuations in groundwater are typically expected to strongly influence GPP (Madani et al., 2020; Palmer et al., 2023). However, this counterintuitive behaviour is consistent with the model implementation of forced saturated groundwater soil levels that prevents any water limitation. A future research avenue could be to make tree root water uptake in the model implementation more sensitive to water deficits.

The response of crop GPP to interannual rainfall variability underscored the critical importance of the timing and distribution of rainfall anomalies during the growing season. Unlike tree GPP, which remains unaffected due to its modeled reliance on

groundwater access, crop GPP is directly influenced by rainfall patterns, reflecting the shallow rooting systems and limited access to deeper water reserves. Crop GPP was highly sensitive to early-season rainfall variability, with deficits, such as in 2019, exposing crops to water shortages during critical growth stages, while surpluses, as in 2020, highlight the benefits of sufficient water availability during early development (Camberlin and Diop, 2003; Zhang et al., 2018). Despite a severe late-season rainfall deficit in 2021, crop GPP reached a similar level as in 2020, suggesting highlighting the interplay between early-season rainfall surpluses and late-season rainfall deficits (Figs. 7 and 8). This suggests that early-season water availability can buffer the effects of subsequent deficits, supporting crop growth and productivity during periods of water stress (Faye et al., 2018; Tovignan et al., 2016).

Given the link between GPP and energy fluxes, as expected, the simulated dry-season energy fluxes remained insensitive to interannual variations in soil water content in the capillary fringe of the groundwater table (Fig. 6). Conversely, rainfall variability directly affected energy fluxes, with wet season anomalies driven by interactions between rainfall and crop GPP (Fig. 7). Notably, when both rainfall and soil water content in the capillary fringe of the groundwater table were variable, the combined effects revealed amplified energy flux anomalies (Fig. 8). Since vegetation modulates energy flux responses (Cicuéndez et al., 2023; Williams and Torn, 2015), efforts must be made to improve the representation of soil-water-vegetation interactions in the model. Indeed, the simulated insensitivity of tree productivity to soil water content variability in this study could limit the model's ability to accurately capture energy flux dynamics and the impacts of interannual hydrological variations in semi-arid ecosystems.

**5 Conclusion**

For this study, a configuration of the ORCHIDEE model was developed to account for the dependence on groundwater of a semi-arid agroforestry system dominated by the reverse phenology and phreatophyte tree *Faidherbia albida*. This new ORCHIDEE configuration was used to study the effect of varying tree density and water availability dynamics (rainfall and soil water content in the capillary fringe of the groundwater table) on carbon and energy fluxes. The ORCHIDEE model demonstrated its capacity to simulate seasonal dynamics of LAI and the annual budget of GPP. Interestingly, the reverse phenology tree (growing during the dry season) was found to contribute significantly to the annual GPP, with around 32 % to 50 % occurring during the dry season. The results of the simulation experiment with varying tree density underscored the critical role of tree density in managing semi-arid ecosystems. Additionally, the analysis of ecosystem sensitivity to interannual water variability underscored the critical importance of rainfall timing and distribution during the wet season for crop productivity. The study also highlighted the need to enhance the responsiveness of tree root water uptake to water variability in the model. Such an adjustment may contribute to a more accurate representation of the impact of soil water content on Faidherbia carbon fluxes and their subsequent influence on energy flux partitioning.

This finding highlights the importance of incorporating the effects of deep-rooted phreatophytic tree species like *Faidherbia albida* in LSMs to accurately represent carbon and energy fluxes in semi-arid regions. Overall, this study emphasizes the importance of accurately representing photosynthetic characteristics, vegetation maps, groundwater dynamics and belowground processes in LSMs to improve simulations in semi-arid ecosystems. Numerical models can inform sustainable land management practices and contribute to the broader understanding of semi-arid ecosystems. Several experimental, monitoring and simulation studies on the Faidherbia-Flux site are currently in progress, and are expected to provide a better understanding of the ecosystem and ecophysiology of *Faidherbia albida*, which could help to further develop the version of ORCHIDEE presented in this study.

## 6 Code and data availability

The ORCHIDEE model (r7949) used in this study is open-source and licensed under the CeCILL (CEA CNRS INRIA Logiciel Libre) license. The model code is archived on Zenodo (https://zenodo.org/records/15313734) (Gaglo et al., 2025b), and installation and usage instructions are provided at https://forge.ipsl.jussieu.fr/orchidee/wiki/Documentation/UserGuide. All climate input data, model outputs, in-situ observations, and analysis scripts used in this study are publicly available on Zenodo at https://doi.org/10.5281/zenodo.17400506 (Gaglo et al., 2025a).

## 7 Author contributions

EG, EC, AV, OR and SL conceptualized and designed the study. OR, CJ, SS, NV and FD provided all necessary data. EG, EC and AV developed the ORCHIDEE Faidherbia configuration code, and EG performed the simulations and the data analysis. EG prepared the paper with contributions from all co-authors.

## 8 Competing interests

The authors declare that they have no conflict of interest.

## 9 Acknowledgments

We thank the students and technicians running the "Faidherbia-Flux" platform (https://lped.info/wikiObsSN/?Faidherbia-Flux) and especially Clémence Chenu, Ablaye Diouf, Ibou Diouf, and Robert Diatte. The "Faidherbia-Flux" research infrastructure and platform are funded by CIRAD, IRD, UMR Eco&Sols, LMI IESOL, ISRA, CERAAS. We gratefully acknowledge the use of the Obelix computing cluster from IPSL, France, which provided access to ORCHIDEE and the essential computing infrastructure that enabled the research presented in this paper. This work was performed using HPC resources from GENCi-

TGCC on grant 2024-06328. Additionally, the authors thank Philippe Peylin, Catherine Ottle, Agnès Ducharne and Josefine Ghattas for sharing their knowledge of the model.

## 10 Financial supports

This work was publicly funded by the DM-TropAFS project through ANR (the French National Research Agency) under the "Investissements d'avenir" programme with the reference ANR-10-LABX-001-01 Labex Agro and coordinated by Agropolis Fondation. Espoir Gaglo gratefully thanks the German Academic Exchange Service (DAAD) for financial support through a scholarship (In Region Scholarship Programme - CERAAS Senegal / 57504744) and CIRAD for providing additional resources to support visits to Eco&Sols through the Action Incitative (AI) Fellowship. Several other projects also contributed to supporting the Faidherbia-Flux platform and the PhD of Espoir Koudjo Gaglo, namely: RAMSES II (EU); CASSECS (EU); SUSTAIN-SAHEL (EU); GALILEO (EU), PEPR FairCarboN (ANR).

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
