# Peer review of "Sensitivity of a Sahelian groundwater-based agroforestry system to tree density and water availability using the land surface model ORCHIDEE (r7949)"

_EGUsphere, 2025_

## Author Comment (AC1)

**Authors' response to rewiever#1**

Manuscript No.: egusphere-2025-1102, submitted to GMD

**Title:** Sensitivity of a Sahelian groundwater-based agroforestry system to tree density and water availability using the land surface model ORCHIDEE (r7949)

**Authors:** Gaglo, K. E., Chaste, E., Luyssaert, S., Roupsard, O., Jourdan, C., Sow, S., Vandewalle, N., Do, F., Ngom, D., and Valade A.

\_\_\_\_\_\_

**Comment #1**

This is a review for the manuscript "Sensitivity of a Sahelian groundwater-based agroforestry system to tree density and water availability using the land surface model ORCHIDEE (r7949)" submitted to Geoscientific Model Development by Gaglo et al. In this work, the authors introduce adjustments made to the ORCHIDEE model in order to simulate a PFT based on a dominant tree species in the Sahel region. The performance of the model is then compared against local flux measurements as well as tested with various scenarios to determine how sensitive it is to different conditions.

Overall, I was quite satisfied with the manuscript, as evidenced by the low number of line-by-line comments. It is a straightforward manuscript that focuses on a relatively simple implementation of a new model version with intentionally limited regional scope. Especially the calibration approach applied here is rather simplistic and carries its own effects that are not truly examined in the manuscript. However, since the examined system is in a semi-arid region that does not receive the appropriate amount of attention when discussing land surface modelling and touches on the various dynamics specific to these areas, I do consider it worthwhile contribution to the ecosystem model development discussion. As a sidenote, my apologies for that horror of a sentence there.

Now I am tempted to recommend acceptance with minor revisions as majority of the work here is easy to follow and comprehend, but the first section of the Discussion, specifically the realism/generality dichotomy here, makes me hesitate. I do comprehend the central idea of the matter and agree it is an important consideration when discussing model development. However, within the context of the work here, it came across as out of place considering that there isn't enough experimentation here to ground majority of the claims there. For example, the argument of limited applicability at other locations is hindered by the fact that there is no experimentation how this PFT performs at those places even with simplified assumptions compared to using the existing PFTs that are not set for environments like this. And that is not even going into how much assumptions already exist in the general ORCHIDEE soil moisture implementation, so using that as a generalist comparison is debatable in itself.

My suggestion is to just remove the majority of the first Discussion section and focus completely on what you have shown here and be more concrete in explaining what the challenges in the larger implementation of the model are here. I understand that is partially the attempt here, but this is muddled by the chosen realism/generality/accuracy approach. Because of this I do think my recommendation is technically return for major submissions, but I do think it should be a relatively minor rewrite here.

**Response #1**

We thank the reviewer for the positive assessment of our manuscript and we appreciate the constructive comment regarding the first section of the discussion section. Please see our detailed point-by-point responses below. Reviewer comments and authors response are in black regular font and all revised manuscript text in bold with modified sections in blue.

**Comment #2**

Line 66: "In magnitude, the water use of Faidherbia albida trees at the plot scale was estimated to be less than 10 % of the annual amount of rainfall (Roupsard et al., 1999; Sarr et al., 2023). However, stable isotope tracing suggested a strong dependence of tree water use on groundwater (Roupsard et al., 1999)."

I was just a little bit confused by this pair of sentences. Is the argument here that the Faidherbia trees use little water, but also draw it from the deeper layers? As the current wording almost implies that the trees use approximately 10 % of the rain fall and in addition draw water from the deeper layers.

Either way, clarify the message here a bit.

**Response #2**

We thank the reviewer for pointing out this ambiguity. Our intention was to highlight that the overall water use of *Faidherbia albida* is relatively small at the plot scale, while the source of this water is largely groundwater rather than rainfall infiltration. To clarify this point, we have revised the sentences as follows:

Revised version lines 66 to 68

At the plot scale, the annual water use of *Faidherbia albida* trees was estimated to represent less than 10% of the rainfall input (Roupsard et al., 1999; Sarr et al., 2023). This modest total use was shown to rely strongly on groundwater by stable isotope tracing (Roupsard et al., 1999).

**Comment #3**

Line 366: "...with an r2 between daily tree LAI simulation and observation of 0.81..."

Just to confirm that the correlation squared value was 0.81 between the observations and simulations? Which would indicate that the correlation itself was 0.9?

Nothing wrong with that, it is simply a staggeringly good fit. What caused me hesitations is that a few lines down the maximum measured LAI is also listed 0.81, which caused me a bit of confusion initially. Coincidences happen but still wished to check. Especially because that is a really high correlation to get when using the switch on phenological approach you seem to be using according to Figure 3.

**Response #3**

We thank the reviewer for his attention to details. Indeed, this is a coincidence, the maximum observed and simulated LAI is 0.8084 and 0.7331 respectively. We highlight that the calibration and validation are performed on different years but one single site explaining the high performance of the calibrated configuration. To prevent readers from being confused about this coincidence of numbers, we rephrased this sentence by only using the RMSE.

Revised version lines 385 to 386

For trees, the 'reverse phenology' of *Faidherbia albida* with its growing season in the dry season was reproduced by the model with an RMSE of 0.15.

\_\_\_\_\_

**Comment #4**

Line 369: "However, 2 out of 3 years..." -> "However, during 2 out of the 3 evaluation years..."

Just to ease the reading a bit, although now I am wondering if I got the preposition right there.

**Response #4**

We thank the reviewer for this suggestion. We agree that adding "during" improves the readability, and the preposition is indeed correct. The revised sentence now reads:

**Revised version lines 388 to 389**

However, during 2 out of the 3 evaluation years, the maximum tree LAI was overestimated by the model by 11% and 13% in 2020–2021 and 2022–2023, respectively (Fig. 3e).

**Comment #5**

Line 570: "As in our case study, their comprehensive evaluation of 13 models revealed that improving one dimension often compromises another, underscoring the difficulty of achieving optimal performance across all three."

This is connected to my general comment, but the discussion preceding this does not really establish anything indicated here.

**Response #5**

We thank the reviewer for this insightful observation. We agree that, despite our intention to situate our study within the broader model evaluation framework of Levins (1966) and subsequent works (e.g. Mahnken et al., 2022), the proposed framing around realism, generality, and accuracy introduced a level of abstraction not fully supported by the scope of our experiments. We also agree with the reviewer that this section risked diverting attention from the concrete results and limitations demonstrated in our analysis.

To address this, we revised Section 4.1 following the reviewer's suggestions by removing the theoretical realism/generality/accuracy arguments and instead highlighting the specific strengths and limitations of our configuration. In particular, we now focus more directly on (i) the limited capture of interannual LAI and day to day GPP variability, (ii) the site-specific calibration that constrains transferability, and (iii) the challenges for scaling this configuration to other regions. The revised text now reads:

Revised version lines 544 to 575

A configuration of the ORCHIDEE r7949 model was developed to represent the phreatophytic behavior of Faidherbia albida trees in Sahelian agroforestry systems. Model evaluation was conducted by comparing simulated LAI, carbon, and energy fluxes with site observations, a standard approach for assessing model performance (Collins et al., 2011; Li et al., 2016; Vuichard et al., 2019).

The model reproduced average LAI values reasonably well for both trees and crops, though interannual variability was not fully captured. For trees, simulated maximum LAI matched measured values but failed to reflect the observed year-to-year fluctuations (Fig. 3a and 3e). This limitation is primarily due to lack of realism in the allocation of carbon in this ORCHIDEE's configuration, which

subsequently limits the interannual plasticity of leaf biomass. Additionally, simplified tree phenology (MacBean et al., 2015) represents a secondary contributing factor. Similarly, crop LAI dynamics compared well with simulations from the process-based model LandscapeDNDC (Rahimi et al., 2021), yet interannual variability remained underrepresented, primarily due to lack of realism in the allocation of carbon in the model, alongside simplified crop phenology and the use of a single crop PFT without accounting for species rotation.

Eddy-covariance-derived GPP is inferred from empirical models fitted to the net ecosystem exchange (NEE) rather than measured directly. Only NEE represents the full semi-hourly variability of the flux, whereas GPP and ecosystem respiration (Reco) are estimated from moving-window fits of light-and temperature-response functions (Lasslop et al., 2010; Reichstein et al., 2012). This procedure inherently smooths short-term fluctuations, leading to a reduction of day-to-day variability in the partitioned components. Unlike EC-derived GPP, the simulated GPP shown stronger day-to-day variability, particularly during the dry season (Fig. 3c and 3g). This reflects the role of vapor pressure deficit (VPD) computed from the meteorological forcing (air temperature, specific humidity and surface pressure; Fig. S2) which limits assimilation via the stomatal scheme (Yin and Struik, 2009). Consequently, short-term changes in atmospheric dryness are transmitted directly to GPP (Fig. S3). The effect is stronger for trees than for crops, consistent with higher stomatal sensitivity in the tree parameterization.

The introduced configuration increased the process realism of the model by explicitly linking tree phenology to groundwater availability, in line with observations of Faidherbia albida water use (Roupsard et al., 1999; Sarr et al., 2023). At the same time, this came at the cost of reduced transferability. The use of exogenous groundwater forcing and parameter calibration based on three years of data at a single site limits the broader applicability of this setup. For example, the current approach is unlikely to capture crop dynamics in regions with different rainfall regimes, soil conditions, or crop management practices. Similarly, the assumption that trees rely exclusively on groundwater omits the possibility of mixed water sources, which may help explain the interannual variability of LAI observed at the site. The configuration improves representation of phreatophyte-driven phenology in ORCHIDEE at the case study site, but its wider applicability remains constrained by data availability and site-specific calibration. Extending the approach to other contexts will require the integration of a dynamic groundwater module coupled with surface water processes, as well as crop and species-specific parameterizations that account for local management and phenological diversity.

---

## Author Comment (AC2)

**Authors' response to rewiever#2**

Manuscript No.: egusphere-2025-1102, submitted to GMD

**Title:** Sensitivity of a Sahelian groundwater-based agroforestry system to tree density and water availability using the land surface model ORCHIDEE (r7949)

**Authors:** Gaglo, K. E., Chaste, E., Luyssaert, S., Roupsard, O., Jourdan, C., Sow, S., Vandewalle, N., Do, F., Ngom, D., and Valade A.

**Comment #1**

This contribution presents the evaluation of the ORCHIDEE model on an agroforestry system in the Sahelian zone, characterized by Faidherbia trees, showing inverted (i.e. leafy in the dry season) phenology. The model is further used to conduct sensitivity analyses of (1) the ecosystem productivity to tree density and (2) year-to-year anomalies of water content in the non-saturated (tapped by the crop) and the saturated zone (tapped by deep-rooted Faidherbia).

I acknowledge the effort of the authors to adapt the ORCHIDEE model to this Sahelian agroforestry system, and evaluate it against LAI and carbon, water and energy flux data. The sensivity analyses are also nicely conducted.

Such a study is classical in its form but all the more informative than developed over a largely understudied ecosystem. More studies of this kind are needed to document the current behaviour (and project the future) of tropical ecosystems.

I see no major flaw in the science and propose the manuscript for acceptance with minor revisions (see my comments below).

**Response #1**

We sincerely thank the reviewer for the positive and constructive assessments and for the valuable suggestions that improved the quality of the manuscript. We revised the manuscript based on the reviewer's comments. Reviewer comments and authors response are in black regular font and all revised manuscript text in bold with modified sections in blue.

\_\_\_\_\_

**Comment #2**

L17: MgC or tC: choose one and stick to it throughout the text

**Response #2**

We thank the reviewer for this helpful observation. We revised the manuscript to ensure consistency by using tC throughout the text instead of MgC. The revised sentence about the annual sequestration rate is now expressed as 0.4 tC ha-1 yr-1 instead of 0.4 MgC ha-1 yr-1.

**Comment #3**

L18: "558 TgC" is expressed in units of stock. I suppose you're right, but please double check you did not mean a flux (in TgC/yr).

**Response #3**

We appreciate the reviewer's attention to this clarification. We verified the source (Luedeling and Neufeldt, 2012) and confirm that the value of 558 TgC refers to a stock difference between treeless croplands and maximum parkland scenarios, not a flux.

**Reference:**

Luedeling, E. and Neufeldt, H.: Carbon sequestration potential of parkland agroforestry in the Sahel, Climatic Change, 115, 443–461, https://doi.org/10.1007/s10584-012-0438-0, 2012.

To avoid ambiguity, we revised the sentence to read:

Revised version lines 16 to 19

The Sahel region is characterized by its semi-arid climate and open-canopy agroforestry systems, which play an important role in global carbon dynamics. Parkland agroforestry has the potential to sequester carbon at an average rate of 0.4 tC ha-1 yr-1, which, if expanded to its maximum potential extent, would correspond to an additional carbon stock of approximately 558 TgC compared to treeless croplands.

**Comment #4**

L107: "pearl millet and groundnut": is it possible to precise the species? (with latin name)

**Response #4**

We thank the reviewer for this suggestion. We revised the text to provide the species names and the varieties used at the site. The revised sentence now reads:

Revised version lines 106 to 107

The main crops are pearl millet (*Pennisetum glaucum* (L.) R. Br., var. Souna) and groundnut (*Arachis hypogaea* L., var. 55–437), conducted in annual rotation.

**Comment #5**

L128-138: Overall clear description of EC data processing, but the method to partition GPP between tree and crop is missing. Please explain.

**Response #5**

We thank the reviewer for this comment. We clarified the text to explain the partitioning method. The revised sentence now reads:

Revised version lines 139 to 142

The partitioning of GPP between tree and crop takes advantage of the reverse phenology of Faidherbia albida: trees are completely defoliated during the rainy season, when crops are growing and are leafy during the dry season, when crops are absent. A simple temporal separation is therefore sufficient to distinguish tree and crop contributions.

**Comment #6**

L177: unsure "physiognomy" is the proper word. Maybe "plant type" or "morphology"?

**Response #6**

We thank the reviewer for this suggestion. We agree that "physiognomy" may be less clear for readers. We revised the text to use "morphology" instead which is more understandable. The revised sentence become:

Revised version lines 180 to 183

Vegetated areas in the model are defined by up to 14 plant functional types (PFTs) in addition to bare soil. A PFT is characterized by a classification scheme that takes into account morphology (tree or grass), leaf morphology (needleleaf or broadleaf), phenology (evergreen, summer-green, or raingreen), photosynthetic pathway (C3 and C4), and climatic zones (boreal, temperate and tropical) (Krinner et al., 2005; Poulter et al., 2015).

**Comment #7**

L182-184: missing is a description of how plants compete for light and water in this configuration of the model.

**Response #7**

Our description of the processes included in ORCHIDEE was indeed too minimal regarding the competition for light and water in the model. We thank the reviewer for giving us the opportunity to clarify this point. The current version of ORCHIDEE includes competition for water between plants of the same morphology (short or tall) but no competition for light between PFTs. Although this is a limitation for applying the model to most agroforestry systems, in the Faidherbia parklands, as mentioned in the introduction (lines 75-77) we believe this is an acceptable configuration given the phreatophyte tree behavior, reverse phenology and low tree density highly limit the competition between trees and crops.

Revised version lines 180 to 190

Vegetated areas in the model are defined by up to 14 plant functional types (PFTs) in addition to bare soil. A PFT is characterized by a classification scheme that takes into account morphology (tree or grass), leaf morphology (needleleaf or broadleaf), phenology (evergreen, summer-green, or raingreen), photosynthetic pathway (C3 and C4), and climatic zones (boreal, temperate and tropical) (Krinner et al., 2005; Poulter et al., 2015). Each PFT is assigned a fraction of the pixel area (the sum of all PFT fractions being 1).

The soil water column is divided into three hydrological tiles with homogeneous soil hydrological properties (Boucher et al., 2020), one for bare soil, one for short vegetation, i.e. croplands and grasslands, and one for tall vegetation, i.e. trees. Within each soil tile all PFTs share the same water, inducing competition, whereas there is no interaction between the water consumption of vegetation in different soil tiles. The energy budget is calculated for each PFT independently and then averaged according to the area fraction of each PFT into a pixel-level energy budget. There is therefore no competition for light between different PFTs.

**Comment #8**

Is Equation 2 calculated on a daily basis (or finer time scale)? Because Wi is dynamic so eq. 2 will modify Rf at the time step of Wi variations, with implication on carbon allocation for the tree.

**Response #8**

We thank the reviewer for noticing a gap in our model description. Equation 2 calculation is on daily basis and all our modifications of water uptake refers to the functional root profile component of the model. The functional root profile defines the soil layers available for plant water uptake and is independent of the root mass present in each layer defined as the structural root profile. A paragraph was added to explain this distinction.

Revised version lines 211 to 213

In ORCHIDEE, the structural root profile defines the root biomass distribution in the soil and hence depends on the carbon allocation. In turn, the functional root profile defines which soil layers plants can draw water from. As only the functional profile was modified in this configuration, root profile hereafter refers to this functional profile, which is defined following Eq. (1).

**Comment #9**

From L212, I understand there is no (i.e. zero) tree water uptake above 4 m? Is it coherent with isotopic analyses of water uptake by Faidherbia (Roupsard et al. cited above)?

**Response #9**

The reviewer's understanding is correct. In the model configuration the tree water uptake is constrained to the soil depths between 4 and 7m. Isotopic observations reported by Roupsard et al. (1999) indicate that the water-table is the dominant source of water for Faidherbia trees except during early rain events when water from superficial soil layers is also absorbed. Attempts to account for this dual water uptake behavior failed to represent the trees' phenology leading us to make the assumption of a pure phreatophyte behavior. We believe this assumption to be credible within the model's framework and discuss its limitations in the discussion section (lines 617-622). Acknowledging the confusion it created for the reviewer, we clarified this assumption in the methods section as well.

Revised version lines 219 to 224

In the Faidherbia configuration, crop and tree root depths were increased to 2 m and 7 m respectively to match field observations and the root profile was adjusted to be partially consistent with recent observations of Siegwart et al., (2023). It was assumed that crops use water from soil layers up to 2 m in depth, whereas tree roots take up water from depths below 4 m (Fig. 2). Tree water uptake from superficial roots was ignored, an assumption supported by their observed low contribution to total tree water use (Roupsard et al., 1999) and is further explained in the discussion section (4.1 and 4.2).

**Comment #10**

L231-232: unclear to me, please rephrase.

**Response #10**

We thank the reviewer for pointing this out. The sentence has been rephrased for clarity to better convey the model logic and the rationale for the rainfall threshold. In the revised text, we emphasize that leaf onset

in our study area is primarily driven by rainfall, and that a minimum precipitation amount is required to avoid triggering leaf growth from very minor rainfall events which can impact water stress variable in the model. We added a sentence to emphasize this explanation:

Revised version lines 241 to 244

In the Faidherbia configuration, the same temperature and water stress conditions was applied but a rainfall amount threshold was added, specifying that leaf onset can only occur if there is a minimum of 10 mm of rainfall over three days (Berg et al., 2010; Marteau et al., 2011; Ndiaye et al., 2024). This constraint ensures that leaf emergence is not triggered by very minor precipitation events.

**Comment #11**

L250-ff (from "In contrast, Jmax..."): sentence unclear or wrong. Rephrase.

**Response #11**

We thank the reviewer for noticing this typo. The sentence was corrected.

Revised version lines 261 to 262

In contrast, Jmax is calculated as a linear function of Vcmax and growth temperature (Kattge and Knorr, 2007).

**Comment #12**

L277 ("the ratio"): according to next sentence, the ratio is insitu/CRUJRA, so please rephrase to "the ratio of the sum of monthly rainfall in the observed data to monthly rainfall in CRUJRA".

**Response #12**

We thank the reviewer for this comment. The sentence has been rephrased to clarify the definition of the ratio and to avoid any ambiguity.

Revised version lines 289 to 290

The ratio of the sum of monthly rainfall in the observed data to the sum of monthly rainfall in CRUJRA was calculated.

**Comment #13**

L370: What are the determinants of LAImax in this version of ORCHIDEE?

**Response #13**

We thank the reviewer for raising this important point. In ORCHIDEE, LAImax is not prescribed as a fixed parameter (as in many models), but emerges from the allocation scheme. Leaf biomass is determined by sapwood mass, root biomass, tree height, and their associated turnover rates, together with carbon allocation rules. These interacting processes ultimately control the maximum LAI that the model produces. We agree that the equation presented in the manuscript (conversion of leaf mass to LAI) only describes how leaf biomass is translated into LAI, not how the maximum is determined. To avoid confusion, we clarified this point in the revised model calibration section:

Revised version lines 333 to 335

The leaf-to-sapwood area ratio was adjusted for LAI until local minimization of RMSE and an acceptable maximum LAI value was reached (Table S1). In this ORCHIDEE configuration, maximum LAI is determined prognostically by the carbon allocation scheme rather than being set as a prescribed parameter. Consequently, we calibrated the leaf-to-sapwood area ratio to serve as a structural constraint to govern the simulated maximum LAI.

**Comment #14**

L358: why not a full 12-month period? (jul-sep missing)

L356-361: the rationale behind definition of periods for the different variables of interest is not clear.

**Response #14**

We apologize for the confusion created by these lines. The time period of study is split between rainy and dry seasons. July-September corresponds to one rainy season that is not part of the analysis described here. We agree with the reviewer that the term "period" has artificially added complexity to the description of the analysis protocol and we have simplified this paragraph by replacing the term "period" with "rainy season" or "dry season".

Revised version lines 376 to 381

The anomalies in SWCC, GPP, LE and H for Faidherbia trees (scenario  $R_{avg}SWCC_{var}$ ), were analyzed over the dry season, that is, from the beginning of October in year n to the end of June in year n+1. In contrast the anomalies in rainfall, GPP, LE and H for crops (scenario  $R_{avg}SWCC_{var}$ ) were analysed over the rainy season, that is, from the beginning of July to the end of September of the same year. Given the variability of both rainfall and SWCC in the scenario  $R_{var}SWCC_{var}$ , anomalies in GPP, LE, and H at ecosystem level were assessed over annual periods covering both dry and rainy seasons.

**Comment #15**

Figure 3: What is rather surprising for tree GPP (and to a lesser extent for crop GPP) is that day-to-day variability in EC-derived GPP is lower than in simulated GPP. Usually, day-to-day variations of GPP are caused by radiation. Is it the case in simulated GPP? how comes this is not the case in EC-derived GPP?

**Response #15**

**1- EC-derived GPP**

We thank the reviewer for this insightful comment. We agree that the lower day-to-day variability of EC-derived GPP compared to simulated GPP may seem counterintuitive, as radiation is usually the dominant driver of short-term GPP variability. However, the EC-derived GPP is not directly measured but estimated from a double fitted model (here the Lasslop et al., 2010 model, eq. 1) for light-response curves and for respiration.

NEE =
$$\frac{\alpha \beta R_g}{\alpha R_g + \beta}$$

+  $rb \exp \left( E_0 \left( \frac{1}{T_{ref} - T_0} - \frac{1}{T_{air} - T_0} \right) \right)$ .

Eq. 1: copy of Eq. 3 from the Lasslop et al., 2010

The light response curve model is the first member at the right side of Eq. 1, the second member gives the model for Ecosystem Respiration (Reco)

Only Net Ecosystem Exchange (NEE) (=Fc) is truly measured by EC. Although NEE truly holds the total semi-hour variability of measured fluxes, GPP and Reco do not, because :

- GPP is fitted on a gliding window of the first member at the right side of Eq. 1
- Reco is obtained from an Arrhenius function (right member of Eq. 1) which is fitted over a sliding window of nighttime NEE measurements (Eq. 2). The semi-hour diurnal Reco is then estimated using the semi-hour daytime temperature, using Eq. 2 with the same parameters than those fitted during the nighttime.

The true variability of daytime GPP and Reco is therefore largely lost and this reduction of variability propagates into the daily sums.

**2- Simulated GPP**

We appreciate the reviewer's insightful comment. The simulated GPP exhibited pronounced day-to-day variability, particularly evident in tree ecosystems. To visualize day-to-day co-variation, time-series of daily GPP and vapor pressure deficit (VPD) were produced. The result shows that daily GPP varied with VPD (strongly during the dry season than the rainy season) which is derived from the meteorological forcing data (air temperature, specific humidity, and surface pressure) (see Fig. S2 for forcing variability; Fig. S3 for diagnostics).

To account for this fair remark about EC-derived GPP and simulated GPP, we added in the manuscript the coefficient of variation of each variable and provided some clarification in the revised manuscript:

Results: Section 3.1.2 (Revised version lines 414 to 416)

Day-to-day variability in EC-derived GPP was lower than that of simulated GPP. This variability was pronounced for trees (Fig. 3c and 3g) and present but weaker for crops (Fig. 3d and 3h). The coefficient of variation over the whole period for the simulated GPP is 0.6 for trees and 1.6 for crops compared to 0.3 and 1.7 for observed tree and crop GPP.

Discussion: Section 4.1 (Revised version lines 556 to 564)

Eddy-covariance-derived GPP is inferred from empirical models fitted to the net ecosystem exchange (NEE) rather than measured directly. Only NEE represents the full semi-hourly variability of the flux, whereas GPP and ecosystem respiration (Reco) are estimated from moving-window fits of light-and temperature-response functions (Lasslop et al., 2010; Reichstein et al., 2012). This procedure inherently smooths short-term fluctuations, leading to a reduction of day-to-day variability in the partitioned components. Unlike EC-derived GPP, the simulated GPP shown stronger day-to-day

variability, particularly during the dry season (Fig. 3c and 3g). This reflects the role of vapor pressure deficit (VPD) computed from the meteorological forcing (air temperature, specific humidity and surface pressure; Fig. S2) which limits assimilation via the stomatal scheme (Yin and Struik, 2009). Consequently, short-term changes in atmospheric dryness are transmitted directly to GPP (Fig. S3). The effect is stronger for trees than for crops, consistent with higher stomatal sensitivity in the tree parameterization.

**References:**

Lasslop, G., Reichstein, M., Papale, D., Richardson, A. D., Arneth, A., Barr, A., Stoy, P., and Wohlfahrt, G.: Separation of net ecosystem exchange into assimilation and respiration using a light response curve approach: critical issues and global evaluation, Global Change Biology, 16, 187–208, https://doi.org/10.1111/j.1365-2486.2009.02041.x, 2010.

Reichstein, M., Stoy, P. C., Desai, A. R., Lasslop, G., and Richardson, A. D.: Partitioning of Net Fluxes, in: Eddy Covariance: A Practical Guide to Measurement and Data Analysis, edited by: Aubinet, M., Vesala, T., and Papale, D., Springer Netherlands, Dordrecht, 263–289, <a href="https://doi.org/10.1007/978-94-007-2351-1\_9">https://doi.org/10.1007/978-94-007-2351-1\_9</a>, 2012.

Yin, X. and Struik, P. C.: C3 and C4 photosynthesis models: An overview from the perspective of crop modelling, NJAS - Wageningen Journal of Life Sciences, 57, 27–38, https://doi.org/10.1016/j.njas.2009.07.001, 2009.

Figure S3: Daily time series comparing simulated gross primary productivity (GPP) (a: tree, b: crop) with vapor deficit pressure (VPD).

**Comment #16**

Table 2 caption: about "n=24", precise this is a 6-year time series \* 4 conditions (4 tree densities).

**Response #16**

We thank the reviewer for this suggestion. The table caption has been revised to clarify the meaning of "n=24."

Revised version lines 466 to 468

Annual statistical summary of gross primary productivity (GPP in tC ha-1 yr-1), energy fluxes (LE and H in GJ m-2 yr-1), and crop harvest (tC ha-1 yr-1) across tree densities (0, 7, 13, and 26 trees). n = 24 (6 years  $\times$  4 tree density conditions). Values followed by different letters are significantly different.

\_\_\_\_\_

**Comment #17**

Figure 5a: When performing the SA to tree density, is all the surface not occupied by trees occupied by crops? I suppose yes, and this is the reason why in the "zero tree" condition, annual GPP reported here (about 10 tC/ha/yr) is higher than annual simulated GPP reported for crops (mixed with trees) in Table 1. This should be clarified in the MM.

**Response #17**

We thank the reviewer for this observation. We clarify that in the sensitivity analysis (section 2.8) all the area not occupied by trees is indeed assumed to be cultivated with crops. Then we added sentence:

Revised version lines 353 to 354

For the sensitivity analysis to tree density, all areas not occupied by *Faidherbia albida* trees were assumed to be cultivated with crops.

**Comment #18**

Figure 6a: it's unclear to me how comes that SWCC anomalies are not distributed homogeneously along the annual cycle. It's said in the MM that the for computing the anomalies in environmental drivers (e.g. SWCC), the average annual cycle (e.g. of SWCC) was used. If that was the case, I would expect for a given day of year that the average SWCC anomaly is zero. This is apparently not the case. Let's take the example of the last day of Rainy season each year (vertical bar on October 1): the negative anomalies of the first two years are not compensated by the positive anomaly in the 3rd year.

**Response #18**

We agree that the description of the calculation of the average SWCC time series lacks clarity as the supplementary text only describes the average cycle omitting the methods for its derivation.

SWCCavg was not calculated as a simple yearly average cycle, instead some key dates and amplitudes of the SWCC cycle and combined with linear interpolation in between those points resulting in the cycle shown in Fig. S3. Then, sensitivity is quantified as the anomaly calculated as the daily difference of all scenarios with respect to RavgSWCCavg considered as the reference scenario. The Materials and methods section 2.8 was revised as follows to clarify this point:

Revised version lines 366 to 374

For this, the average yearly cycle of precipitation and groundwater dynamics were respectively computed to be used as multi-year time series without any interannual variability. For SWCC, the average cycle was reconstructed from key observed dates and amplitudes of the groundwater fluctuation (1 Jan: 0.26 m³ m⁻³; 8 Jul: 0.15 m³ m⁻³; 10 Sep: 0.15 m³ m⁻³; 24 Oct: 0.31 m³ m⁻³; 31 Dec: 0.26 m³ m⁻³), and linear interpolation was applied between these points to produce a smoothed climatological cycle (Fig. S3). These are called the "average" scenarios (see Text S3 and Fig. S3 for details on the calculation in the Supplement). The sensitivity analysis consisted of four climate and SWCC combinations: i) simulation with average rain and average SWCC (RavgSWCCavg), ii) simulation with average rain and variable SWCC (RavgSWCCvar), iii) simulation with variable rain and average SWCC (RvarSWCCavg) and iv) simulation with variable rain and variable SWCC (RvarSWCCvar). In all simulations, the other climate forcing variables were taken from CRUJRA climate forcing data.

The sensitivity is quantified as the anomaly calculated as the daily difference of all scenarios with respect to  $R_{avg}SWCC_{avg}$  considered as the reference scenario.

The Supplementary Text S3 section was rewritten as follows to clarify this point:

SWCCavg was not calculated as a simple yearly average cycle, instead some key dates and amplitudes of the SWCC cycle and combined with linear interpolation in between those points resulting in the cycle shown in Fig. S3. Indeed, at the beginning of the year (1 January), SWCC averaged 0.26 m³ m⁻³ and gradually declined to its minimum of 0.15 m³ m⁻³ by 8 July, marking the driest period of the year. This low level persisted until 10 September, after which a rapid recharge phase began, reaching its peak value of 0.31 m³ m⁻³ on 24 October. By 31 December, SWCCavg returned to around 0.26 m³ m⁻³, closing the annual cycle. This corresponds to an overall amplitude of 0.16 m³ m⁻³, with a recharge duration of about 31 days. The late-season rise in SWCC occurs near the end of the rainy season, as a delayed but rapid response of the groundwater to rainfall, coinciding with the budburst of Faidherbia albida in mid-October (Roupsard et al., 2022).
* * *
**Comment #19**

L522-523: failure to reproduce IAV of LAI are primarily due to lack of realism in the allocation part of the model (LAImax) rather than in phenological modules.

**Response #19**

We thank the reviewer for this helpful observation. We agree that in this configuration of ORCHIDEE for Faidherbia; the limited interannual variability of LAI is mainly related to the allocation scheme, which stabilizes leaf biomass and constrains year-to-year fluctuations. Simplified phenology may also contribute but plays a secondary role. We have revised the discussion to reflect this point.

Revised version lines 185 to 190

For trees, simulated maximum LAI matched measured values but failed to reflect the observed year-to-year fluctuations (Fig. 3a and 3e). This limitation is primarily due to lack of realism in the allocation of carbon in this ORCHIDEE's configuration, which subsequently limits the interannual plasticity of leaf biomass. Additionally, simplified tree phenology (MacBean et al., 2015) represents a secondary contributing factor. Similarly, crop LAI dynamics compared well with simulations from the process-based model LandscapeDNDC (Rahimi et al., 2021), yet interannual variability remained

underrepresented, primarily due to lack of realism in the allocation of carbon, alongside simplified crop phenology and the use of a single crop PFT without accounting for species rotation.

**Comment #20**

L545: "The model's generality is estimated to have decreased in the developed configuration...". Rephrase to make clear that "the developed configuration" is the model version you are using in this manuscript.

**Response #20**

Another reviewer questioned the relevance of the discussion of the generality/realism/accuracy aspects of the model in the context of this study. We therefore followed their advice and simplified this section of the manuscript by removing these arguments. This sentence has therefore disappeared from the revised manuscript.

**Comment #21**

L588: "competition for space"... and not competition for light?

**Response #21**

As answered to a previous comment, indeed each PFT (in our case deciduous trees and crops) is allocated a fraction of the pixel each with its own energy budget. There is therefore no competition for light. We hope the lines added in the model description section addresses the confusion created here.

Revised version lines 185 to 190

The soil water column is divided into three hydrological tiles with homogeneous soil hydrological properties (Boucher et al., 2020), one for bare soil, one for short vegetation, i.e. croplands and grasslands, and one for tall vegetation, i.e. trees. Within each soil tile all PFTs share the same water, inducing competition, whereas there is no interaction between the water consumption of vegetation in different soil tiles. The energy budget is calculated for each PFT independently and then averaged according to the area fraction of each PFT into a pixel-level energy budget. There is therefore no competition for light between different PFTs.

\_\_\_\_\_

**Comment #22**

L976: I suppose the reference for the paper of Vickers and Mahrt is wrong. Should be Vickers, D., & Mahrt, L. (1997). Quality control and flux sampling problems for tower and aircraft data. Journal of Atmospheric and Oceanic Technology, 14(3), 512-526.

**Response #22**

We thank the reviewer for pointing out the error. The reference has been corrected to the accurate citation:

Revised version lines 954 to 955

Vickers, D. and Mahrt, L.: Quality Control and Flux Sampling Problems for Tower and Aircraft Data, J. Atmospheric Ocean. Technol., 14, 512–526, https://doi.org/10.1175/1520-0426(1997)014%253C0512:QCAFSP%253E2.0.CO;2, 1997.

\_\_\_\_\_